



Atmospheric
Chemistry
and Physics

# Ozone seasonal evolution and photochemical production regime in the polluted troposphere in eastern China derived from high-resolution Fourier transform spectrometry (FTS) observations

Youwen Sun[1], Cheng Liu[1,2,3,4], Mathias Palm[5], Corinne Vigouroux[5], Justus Notholt[5], Qihou Hu[1], Nicholas Jones[7], Wei Wang[1], Wenjing Su[2], Wenqiang Zhang[2], Changong Shan[1], Yuan Tian[1], Xingwei Xu[1], Martine De Mazière[6], Minqiang Zhou[6], and Jianguo Liu[1]

[1]Key Laboratory of Environmental Optics and Technology, Anhui Institute of Optics and Fine Mechanics,
Chinese Academy of Sciences, Hefei 230031, China
[2]School of Earth and Space Sciences, University of Science and Technology of China, Hefei, 230026, China
[3]Center for Excellence in Regional Atmospheric Environment, Institute of Urban Environment,
Chinese Academy of Sciences, Xiamen, 361021, China
[4]Anhui Province Key Laboratory of Polar Environment and Global Change, USTC, Hefei, 230026, China
[5]University of Bremen, Institute of Environmental Physics, P.O. Box 330440, 28334 Bremen, Germany
[6]Royal Belgian Institute for Space Aeronomy (BIRA-IASB), Brussels, Belgium
[7]School of Chemistry, University of Wollongong, Northfields Ave, Wollongong, NSW, 2522, Australia

**Correspondence:** Cheng Liu (chliu81@ustc.edu.cn)

**Abstract.** The seasonal evolution of $O_3$ and its photochemical production regime in a polluted region of eastern China between 2014 and 2017 has been investigated using observations. We used tropospheric ozone ($O_3$), carbon monoxide (CO), and formaldehyde (HCHO, a marker of VOCs (volatile organic compounds)) partial columns derived from high-resolution Fourier transform spectrometry (FTS); tropospheric nitrogen dioxide ($NO_2$, a marker of $NO_x$ (nitrogen oxides)) partial column deduced from the Ozone Monitoring Instrument (OMI); surface meteorological data; and a back trajectory cluster analysis technique. A broad $O_3$ maximum during both spring and summer (MAM/JJA) is observed; the day-to-day variations in MAM/JJA are generally larger than those in autumn and winter (SON/DJF). Tropospheric $O_3$ columns in June are $1.55 \times 10^{18}$ molecules cm$^{-2}$ (56 DU (Dobson units)), and in December they are $1.05 \times 10^{18}$ molecules cm$^{-2}$ (39 DU). Tropospheric $O_3$ columns in June were $\sim 50\%$ higher than those in December. Compared with the SON/DJF season, the observed tropospheric $O_3$ levels in MAM/JJA are more influenced by the transport of air masses from densely populated and industrialized areas, and the high $O_3$ level and variability in MAM/JJA

is determined by the photochemical $O_3$ production. The tropospheric-column HCHO/$NO_2$ ratio is used as a proxy to investigate the photochemical $O_3$ production rate (PO$_3$). The results show that the PO$_3$ is mainly nitrogen oxide ($NO_x$) limited in MAM/JJA, while it is mainly VOC or mixed VOC–$NO_x$ limited in SON/DJF. Statistics show that $NO_x$-limited, mixed VOC–$NO_x$-limited, and VOC-limited PO$_3$ accounts for 60.1%, 28.7%, and 11% of days, respectively. Considering most of PO$_3$ is $NO_x$ limited or mixed VOC–$NO_x$ limited, reductions in $NO_x$ would reduce $O_3$ pollution in eastern China.

## 1 Introduction

Human health, terrestrial ecosystems, and material degradation are impacted by poor air quality resulting from high photochemical ozone ($O_3$) levels (Wennberg and Dabdub, 2008; Edwards et al., 2013; Schroeder et al., 2017). In polluted areas, tropospheric $O_3$ is generated from a series of complex reactions in the presence of sunlight involving carbon monoxide (CO), nitrogen oxides ($NO_x \equiv NO$ (nitric oxide) + $NO_2$

(nitrogen dioxide)), and volatile organic compounds (VOCs) (Oltmans et al., 2006; Schroeder et al., 2017). Briefly, VOCs first react with the hydroxyl radical (OH) to form a peroxy radical ($HO_2 + RO_2$), which increases the rate of catalytic cycling of NO to $NO_2$. $O_3$ is then produced by photolysis of $NO_2$. Subsequent reactions between $HO_2$ or $RO_2$ and NO lead to radical propagation (via subsequent reformation of OH). Radical termination proceeds via the reaction of OH with $NO_x$ to form nitric acid ($HNO_3$) (Reaction R1, referred to as $LNO_x$) or by radical–radical reactions resulting in stable peroxide formation (Reactions R2–R4, referred to as $LRO_x$, where $RO_x \equiv RO_2 + HO_2$) (Schroeder et al., 2017):

$$OH + NO_2 \rightarrow HNO_3, \tag{R1}$$

$$2HO_2 \rightarrow H_2O_2 + O_2, \tag{R2}$$

$$HO_2 + RO_2 \rightarrow ROOH + O_2, \tag{R3}$$

$$2RO_2 \rightarrow ROOR + O_2. \tag{R4}$$

Typically, the relationship between these two competing radical termination processes (referred to as the ratio $LRO_x/LNO_x$) can be used to evaluate the photochemical regime. In high-radical, low-$NO_x$ environments, Reactions (R2)–(R4) remove radicals at a faster rate than Reaction (R1) (i.e., $LRO_x \gg LNO_x$), and the photochemical regime is regarded as "$NO_x$ limited". In low-radical, high-$NO_x$ environments the opposite is true (i.e., $LRO_x \ll LNO_x$), and the regime is regarded as "VOC limited". When the rates of the two loss processes are comparable ($LNO_x \approx LRO_x$), the regime is said to be at the photochemical transition/ambiguous point, i.e., mixed VOC–$NO_x$ limited (Kleinman et al., 2005; Sillman et al., 1995a; Schroeder et al., 2017).

Understanding the photochemical regime at local scales is a crucial piece of information for enacting effective policies to mitigate $O_3$ pollution (Jin et al., 2017; Schroeder et al., 2017). In order to determine the regime, the total reactivity with OH of the myriad of VOCs in the polluted area has to be estimated (Sillman, 1995a; Xing et al., 2017). In the absence of such information, the formaldehyde (HCHO) concentration can be used as a proxy for VOC reactivity because it is a short-lived oxidation product of many VOCs and is positively correlated with peroxy radicals (Schroeder et al., 2017). Sillman (1995a) and Tonnesen and Dennis (2000) found that in situ measurements of the ratio of HCHO (a marker of VOCs) to $NO_2$ (a marker of $NO_x$) could be used to diagnose local photochemical regimes. Over polluted areas, both HCHO and tropospheric $NO_2$ have vertical distributions that are heavily weighted toward the lower troposphere, indicating that tropospheric-column measurements of these gases are fairly representative of near-surface conditions. Many studies have taken advantage of these favorable vertical distributions to investigate surface emissions of $NO_x$ and VOCs from space (Boersma et al., 2009; Martin et al., 2004a; Millet et al., 2008; Streets et al., 2013). Martin et al. (2004a) and Duncan et al. (2010) used satellite measurements of the column $HCHO/NO_2$ ratio to explore tropospheric $O_3$ sensitivities from space and disclosed that this diagnosis of $O_3$ production rate ($PO_3$) is consistent with previous findings of surface photochemistry. Witte et al. (2011) used a similar technique to estimate changes in $PO_3$ from the strict emission control measures (ECMs) during the Beijing Summer Olympic Games period in 2008. Recent papers have applied the findings of Duncan et al. (2010) to observe $O_3$ sensitivity in other parts of the world (Choi et al., 2012; Witte et al., 2011; Jin and Holloway, 2015; Mahajan et al., 2015; Jin et al., 2017).

With in situ measurements, Tonnesen and Dennis (2000) observed a radical-limited environment with $HCHO/NO_2$ ratios $< 0.8$, an $NO_x$-limited environment with $HCHO/NO_2$ ratios $> 1.8$, and a transition environment with $HCHO/NO_2$ ratios between 0.8 and 1.8. With 3-D chemical model simulations, Sillman (1995a) and Martin et al. (2004b) estimated that the transition between the VOC- and $NO_x$-limited regimes occurs when the $HCHO/NO_2$ ratio is $\sim 1.0$. With a combination of regional chemical model simulations and the Ozone Monitoring Instrument (OMI) measurements, Duncan et al. (2010) concluded that $O_3$ production decreases with reductions in VOCs at a column $HCHO/NO_2$ ratio $< 1.0$ and $NO_x$ at column $HCHO/NO_2$ ratio $> 2.0$; both $NO_x$ and VOC reductions decrease $O_3$ production when the column $HCHO/NO_2$ ratio lies in between 1.0 and 2.0. With a 0-D photochemical box model and airborne measurements, Schroeder et al. (2017) presented a thorough analysis of the utility of column $HCHO/NO_2$ ratios to indicate surface $O_3$ sensitivity and found that the transition/ambiguous range estimated via column data is much larger than that indicated by in situ data alone. Furthermore, Schroeder et al. (2017) concluded that many additional sources of uncertainty (regional variability, seasonal variability, variable free-tropospheric contributions, retrieval uncertainty, air pollution levels and meteorological conditions) may cause the transition threshold vary both geographically and temporally, and thus the results from one region are not likely to be applicable globally.

With the rapid increase in fossil fuel consumption in China over the past 3 decades, the emission of chemical precursors of $O_3$ ($NO_x$ and VOCs) has increased dramatically, surpassing that of North America and Europe and raising concerns about worsening $O_3$ pollution in China (Tang et al., 2012; Wang et al., 2017; Xing et al., 2017). Tropospheric $O_3$ was already included in the new air quality standard as a routine monitoring component (http://www.mep.gov.cn; last access: 23 May 2018), where the limit for the maximum daily 8 h average (MDA8) $O_3$ in urban and industrial areas is $160 \, \mu g \, m^{-3}$ ($\sim 75$ ppbv at 273 K, 101.3 kPa). According to air quality data released by the Chinese Ministry of Environmental Protection, tropospheric $O_3$ has replaced $PM_{2.5}$ as the primary pollutant in many cities during summer (http://www.mep.gov.cn/; last access: 23 May 2018). A precise knowledge of $O_3$ evolution and photochemical pro-

duction regime in the polluted troposphere in China has important policy implications for $O_3$ pollution controls (Tang et al., 2011; Xing et al., 2017; Wang et al., 2017).

In this study, we investigate the $O_3$ seasonal evolution and photochemical production regime in the polluted troposphere in eastern China with tropospheric $O_3$, CO, and HCHO derived from ground-based high-resolution Fourier transform spectrometry (FTS) in Hefei, China, tropospheric $NO_2$ deduced from the OMI satellite (https://aura.gsfc.nasa.gov/omi. html; last access: 23 May 2018), surface meteorological data, and a back trajectory cluster analysis technique. Considering the fact that most NDACC (Network for Detection of Atmospheric Composition Change) FTS sites are located in Europe and Northern America, whereas the number of sites in Asia, Africa, and South America is very sparse, and there is still no official NDACC FTS station that covers China (http://www.ndacc.org/; last access: 23 May 2018), this study can not only improve our understanding of regional photochemical $O_3$ production regime but also contributes to the evaluation of $O_3$ pollution controls.

This study concentrates on measurements recorded during midday, when the mixing layer has largely been dissolved. All FTS retrievals are selected within $\pm 30$ min of OMI overpass time (13:30 local time (LT)). While the FTS instrument can measure throughout the whole day, unless cloudy, OMI measures only during midday. For Hefei, this coincidence criterion is a balance between the accuracy and the number of data points.

## 2 Site description and instrumentation

The FTS observation site ($117°10'$ E, $31°54'$ N; 30 m a.s.l. (above sea level)) is located in the western suburbs of Hefei city (the capital of Anhui Province, population of 8 million) in central-eastern China (Fig. S1 in the Supplement). A detailed description of this site and its typical observation scenario can be found in Tian et al. (2018). Similar to other Chinese megacities, serious air pollution is common in Hefei throughout the whole year (http://mep.gov.cn/; last access: 23 May 2018).

Our observation system consists of a high-resolution FTS spectrometer (IFS125HR, Bruker GmbH, Germany), a solar tracker (Tracker-A Solar 547, Bruker GmbH, Germany), and a weather station (ZENO-3200, Coastal Environmental Systems, Inc., USA). The near-infrared (NIR) and middle infrared (MIR) solar spectra were alternately acquired in routine observations (Wang et al., 2017). The MIR spectra used in this study are recorded over a wide spectral range (about $600–4500 \, cm^{-1}$) with a spectral resolution of $0.005 \, cm^{-1}$. The instrument is equipped with a KBr beam splitter and MCT detector for $O_3$ measurements and a KBr beam splitter and InSb detector for other gases. The weather station includes sensors for air pressure ($\pm 0.1$ hpa), air temperature ($\pm 0.3$ °C), relative humidity ($\pm 3$ %), solar radiation ($\pm 5$ %),

wind speed ($\pm 0.2 \, m \, s^{-1}$), wind direction ($\pm 5°$), and the presence of rain.

## 3 FTS retrievals of $O_3$, CO, and HCHO

### 3.1 Retrieval strategy

The SFIT4 (version 0.9.4.4) algorithm is used in the profile retrieval (Supplement Sect. S1; https://www2.acom.ucar. edu/irwg/links; last access: 23 May 2018). The retrieval settings for $O_3$, CO, and HCHO are listed in Table 1. All spectroscopic line parameters are adopted from HITRAN 2008 (Rothman et al., 2009). A priori profiles of all gases except $H_2O$ are from a dedicated WACCM (Whole Atmosphere Community Climate Model) run. A priori profiles of pressure, temperature, and $H_2O$ are interpolated from the National Centers for Environmental Protection and National Center for Atmospheric Research (NCEP/NCAR) reanalysis (Kalnay et al., 1996). For $O_3$ and CO, we follow the NDACC standard convention with respect to micro-window (MW) selection and interfering gas consideration (https://www2. acom.ucar.edu/irwg/links; last access: 23 May 2018). HCHO is not yet an official NDACC species but has been retrieved at a few stations with different retrieval settings (Albrecht et al., 2002; Vigouroux et al., 2009; Jones et al., 2009; Viatte et al., 2014; Franco et al., 2015). The four MWs used in the current study are chosen from a harmonization project taking place in view of future satellite validation (Vigouroux et al., 2018). They are centered at around $2770 \, cm^{-1}$, and the interfering gases are $CH_4$, $O_3$, $N_2O$, and HDO.

We assume measurement noise covariance matrices $\mathbf{S}_\varepsilon$ to be diagonal and set their diagonal elements to the inverse square of the signal-to-noise ratio (SNR) of the fitted spectra and the non-diagonal elements of $\mathbf{S}_a$ [CE1] to zero. For all gases, the diagonal elements of a priori profile covariance matrices $\mathbf{S}_a$ are set to the standard deviation of a dedicated WACCM run from 1980 to 2020, and its non-diagonal elements are set to zero.

We regularly used a low-pressure HBr cell to monitor the instrument line shape (ILS) and included the measured ILS in the retrieval (Hase, 2012; Sun et al., 2018).

### 3.2 Profile information in the FTS retrievals

The sensitive range for CO and HCHO is mainly tropospheric, and for $O_3$ it is both tropospheric and stratospheric (Fig. S2). The typical degrees of freedom (DOFS) over the total atmosphere obtained at Hefei for each gas are included in Table 2: they are about 4.8, 3.5, and 1.2 for $O_3$, CO, and HCHO, respectively. In order to separate independent partial column amounts in the retrieved profiles, we have chosen the altitude limit for each independent layer such that the DOFS in each associated partial column is not less than 1.0. The retrieved profiles of $O_3$, CO, and HCHO can be divided into four, three, and one independent layers, respec-

**Table 1.** Summary of the retrieval parameters used for $O_3$, CO, and HCHO. All micro-windows (MWs) are given cm$^{-1}$.

| Gases | | $O_3$ | CO | HCHO |
|---|---|---|---|---|
| Retrieval code | | SFIT4 v 0.9.4.4 | SFIT4 v 0.9.4.4 | SFIT4 v 0.9.4.4 |
| Spectroscopy | | HITRAN2008 | HITRAN2008 | HITRAN2008 |
| $P$, $T$, $H_2O$ profiles | | NCEP | NCEP | NCEP |
| A priori profiles for target/interfering gases except $H_2O$ | | WACCM | WACCM | WACCM |
| MW for profile retrievals | | 1000–1004.5 | 2057.7–2058 2069.56–2069.76 2157.5–2159.15 | 2763.42–2764.17 2765.65–2766.01 2778.15–2779.1 2780.65–2782.0 |
| Retrieved interfering gases | | $H_2O$, $CO_2$, $C_2H_4$, $^{668}O_3$, $^{686}O_3$ | $O_3$, $N_2O$, $CO_2$, OCS, $H_2O$ | $CH_4$, $O_3$, $N_2O$, HDO |
| SNR for de-weighting | | None | 500 | 600 |
| Regularization | $S_a$ | Diagonal: 20 % No correlation | Diagonal: 11 %–27 % No correlation | Diagonal: 10 % No correlation TS1 |
| ILS | $S_\varepsilon$ | Real SNR LINEFIT145 | Real SNR LINEFIT145 | Real SNR LINEFIT145 |
| Error analysis | | Systematic error – smoothing error (smoothing) – errors from other parameters: background curvature (curvature), optical path difference (max_opd), field of view (omega), solar line strength (solstrnth), background slope (slope), solar line shift (solshft), phase (phase), solar zenith angle(sza), line temperature broadening (linetair_gas), line pressure broadening (linepair_gas), line intensity(lineint_gas) | | |
| | | Random error – interference errors: retrieval parameters (retrieval_parameters), interfering species (interfering_species) – Measurement error (measurement) – errors from other parameters: temperature (temperature), zero level (zshift) | | |

tively (Fig. S3). The troposphere is well resolved by $O_3$, CO, and HCHO, where CO exhibits the best vertical resolution with more than two independent layers in the troposphere.

In this study, we have chosen the same upper limit (12 km) for the tropospheric columns for all gases (Table 2), which is about 3 km lower than the mean value of the tropopause ($\sim$ 15.1 km). In this way we ensured the accuracies for the tropospheric $O_3$, CO, and HCHO retrievals and minimized the influence of transport from the stratosphere, i.e., the so-called STE process (stratosphere–troposphere exchange).

## 3.3 Error analysis

The results of the error analysis presented here are based on the average of all measurements that fulfill the screening scheme, which is used to minimize the impacts of sig-

nificant weather events or instrument problems (Supplement Sect. S2). In the troposphere, the dominant systematic error for $O_3$ and CO is the smoothing error, and for HCHO it is the line intensity error (Fig. S4). The dominant random error for $O_3$ and HCHO is the measurement error, and for CO it is the zero baseline level error (Fig. S5). Taking all error items into account, the summarized errors in $O_3$, CO, and HCHO for the 0–12 km tropospheric partial column and for the total column are listed in Table 3. The total errors in the tropospheric partial columns for $O_3$, CO, and HCHO have been evaluated to be 8.7 %, 6.8 %, and 10.2 %, respectively.

**Table 2.** Typical degrees of freedom for signal (DOFs) and sensitive range of the retrieved $O_3$, CO, and HCHO profiles at Hefei site.

| Gas | Total column DOFs | Sensitive range (km) | Tropospheric partial column (km) | Tropospheric DOFs |
|-----|-----|-----|-----|-----|
| $O_3$ | 4.8 | Ground – 44 | Ground – 12 | 1.3 |
| CO | 3.5 | Ground – 27 | Ground – 12 | 2.7 |
| HCHO | 1.2 | Ground – 18 | Ground – 12 | 1.1 |

**Table 3.** Errors in % of the column amount of $O_3$, CO, and HCHO for the 0–12 km tropospheric partial column and for the total column.

| Gas | $O_3$ | | CO | | HCHO | |
|-----|-----|-----|-----|-----|-----|-----|
| Altitude (km) | 0–12 | Total column | 0–12 | Total column | 0–12 | Total column |
| Total random | 3.2 | 0.59 | 3.8 | 0.66 | 3.3 | 0.97 |
| Total systematic | 8.1 | 4.86 | 5.7 | 3.9 | 9.6 | 5.7 |
| Total errors | 8.7 | 5.0 | 6.8 | 3.95 | 10.2 | 5.8 |

## 4 Tropospheric $O_3$ seasonal evolution

### 4.1 Tropospheric $O_3$ seasonal variability

Figure 1a shows the tropospheric $O_3$ column time series recorded by the FTS from 2014 to 2017, where we followed Gardiner's method and used a second-order Fourier series plus a linear component to determine the annual variability (Gardiner et al., 2008). The analysis did not indicate a significant secular trend of tropospheric $O_3$ column probably because the time series is much shorter than those in Gardiner et al. (2008); the observed seasonal cycle of tropospheric $O_3$ variations is well captured by the bootstrap resampling method (Gardiner et al., 2008). As commonly observed, high levels of tropospheric $O_3$ occur in spring and summer (hereafter MAM/JJA). Low levels of tropospheric $O_3$ occur in autumn and winter (hereafter SON/DJF). Day-to-day variations in MAM/JJA are generally larger than those in SON/DJF (Fig. 1b). At the same time, the tropospheric $O_3$ column roughly increases over time in the first half of the year and reaches the maximum in June and then decreases during the second half of the year. Tropospheric $O_3$ columns in June are $1.55 \times 10^{18}$ molecules cm$^{-2}$ (56 DU (Dobson units)) and in December are $1.05 \times 10^{18}$ molecules cm$^{-2}$ (39 DU). Tropospheric $O_3$ columns in June were $\sim 50\%$ higher than those in December.

Vigouroux et al. (2015) studied the $O_3$ trends and variabilities at eight NDACC FTS stations that have a long-term time series of $O_3$ measurements, namely, Ny-Ålesund (79° N), Thule (77° N), Kiruna (68° N), Harestua (60° N), Jungfraujoch (47° N), Izaña (28° N), Wollongong (34° S), and Lauder (45° S). All these stations were located in non-polluted or relatively clean areas. The tropospheric columns at these stations are of the order of $0.7 \times 10^{18}$ to $1.1 \times$ $10^{18}$ molecules cm$^{-2}$. The results showed a maximum tropospheric $O_3$ column in spring at all these stations except at the high-altitude stations Jungfraujoch and Izaña, where it extended into early summer. This is because the STE process is most effective during late winter and spring (Vigouroux et al., 2015). In contrast, we observed a broader maximum at Hefei which extends over the MAM/JJA season, and the values are $\sim 35\%$ higher than those studied in Vigouroux et al. (2015). This is because the observed tropospheric $O_3$ levels in MAM/JJA are more influenced by air masses originating from densely populated and industrialized areas (see Sect. 4.2), and the MAM/JJA meteorological conditions are more favorable to photochemical $O_3$ production (see Sect. 5.1). The selection of tropospheric limits 3 km below the tropopause minimized but cannot avoid the influence of transport from the stratosphere; the STE process may also contribute to high level of tropospheric $O_3$ column in spring.

### 4.2 Regional contribution to tropospheric $O_3$ levels

In order to determine where the air masses came from and thus contributed to the observed tropospheric $O_3$ levels, we have used the HYSPLIT (Hybrid Single-Particle Lagrangian Integrated Trajectory) model to calculate the three-dimensional kinematic back trajectories that coincide with the FTS measurements from 2014 to 2017 (Draxler et al., 2009). In the calculation, the GDAS (University of Alaska Fairbanks GDAS Archive) meteorological fields were used with a spatial resolution of $0.25° \times 0.25°$, a time resolution of 6 h, and 22 vertical levels from the surface to 250 mbar. All daily back trajectories at 12:00 UTC, with a 24 h pathway arriving at Hefei site at 1500 m a.s.l., have been grouped into clusters and divided into MAM/JJA and SON/DJF seasons (Stunder, 1996). The results showed that air masses in

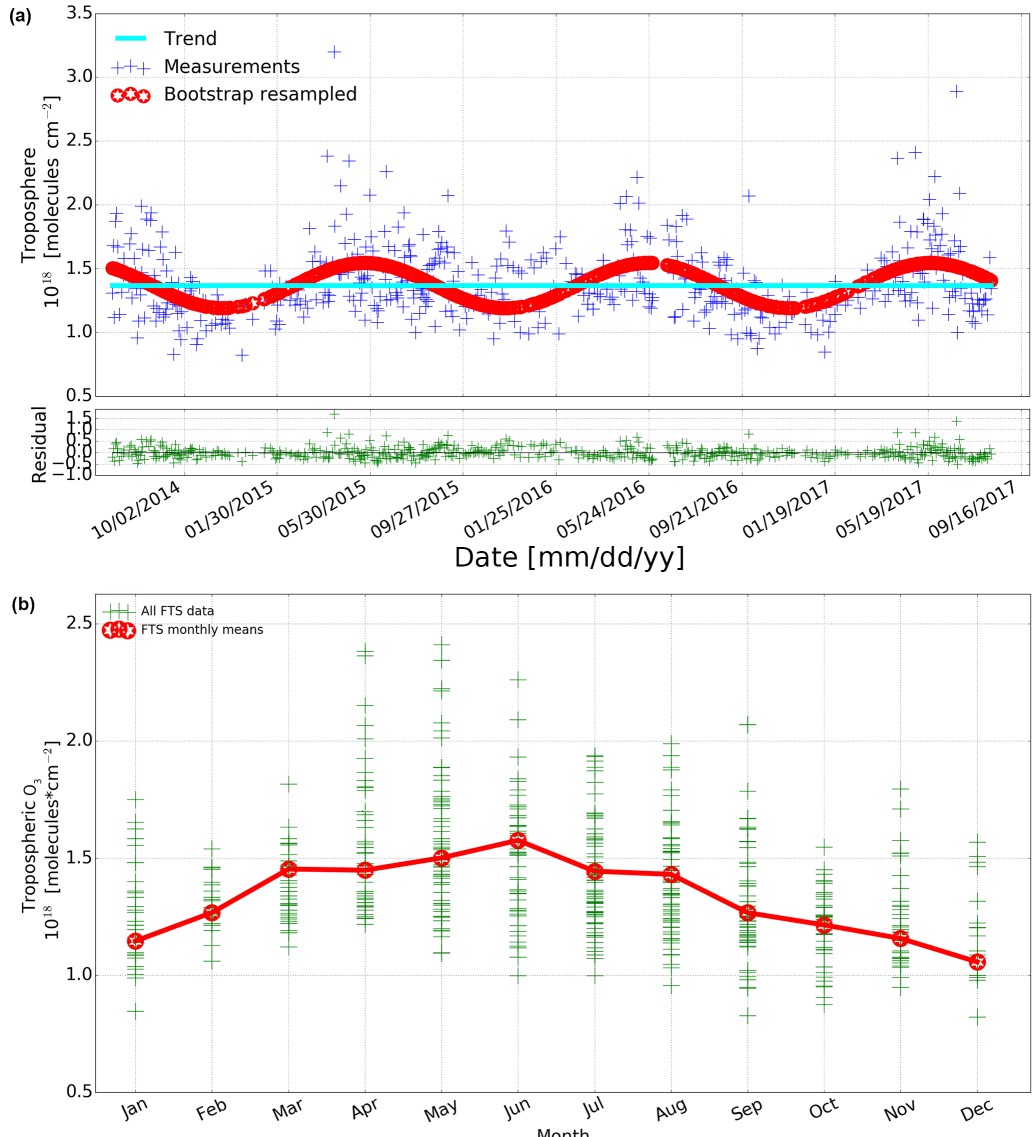

**Figure 1. (a)** FTS measured and bootstrap resampled tropospheric O$_3$ columns at Hefei site. The linear trend and the residual are also shown. Detailed description of the bootstrap method can be found in Gardiner et al. (2008). **(b)** Tropospheric O$_3$ column monthly means derived from **(a)**.

Jiangsu and Anhui provinces in eastern China; Hebei and Shandong provinces in northern China; Shaanxi, Henan, and Shanxi provinces in northwestern China; and Hunan and Hubei provinces in central China contributed to the observed tropospheric O$_3$ levels.

In the MAM/JJA season (Fig. 2a), 28.8 % of air masses are of eastern origin and arrived at Hefei through the southeast of Jiangsu Province and east of Anhui Province; 41.0 % are of southwestern origin and arrived at Hefei through the northeast of Hunan and Hubei provinces, and southwest of Anhui Province; 10.1 % are of northwestern origin and arrived at Hefei through the southeast of Shanxi and Henan provinces, and northwest of Anhui Province; 10.1 % are of

northern origin and arrived at Hefei through the south of Shandong Province and north of Anhui Province; 10.1 % are of local origin generated in the south of Anhui Province. As a result, air pollution from megacities such as Shanghai, Nanjing, Hangzhou, and Hefei in eastern China; Changsha and Wuhan in central-southern China; Zhenzhou and Taiyuan in northwest China; and Jinan in north China could contribute to the observed tropospheric O$_3$ levels.

In the SON/DJF season, trajectories are generally longer and originated in the northwest of the MAM/JJA ones (Fig. 2b). The direction of air masses originating in the eastern sector shifts from the southeast to the northeast of Jiangsu Province, and that of local air masses shifts from the south

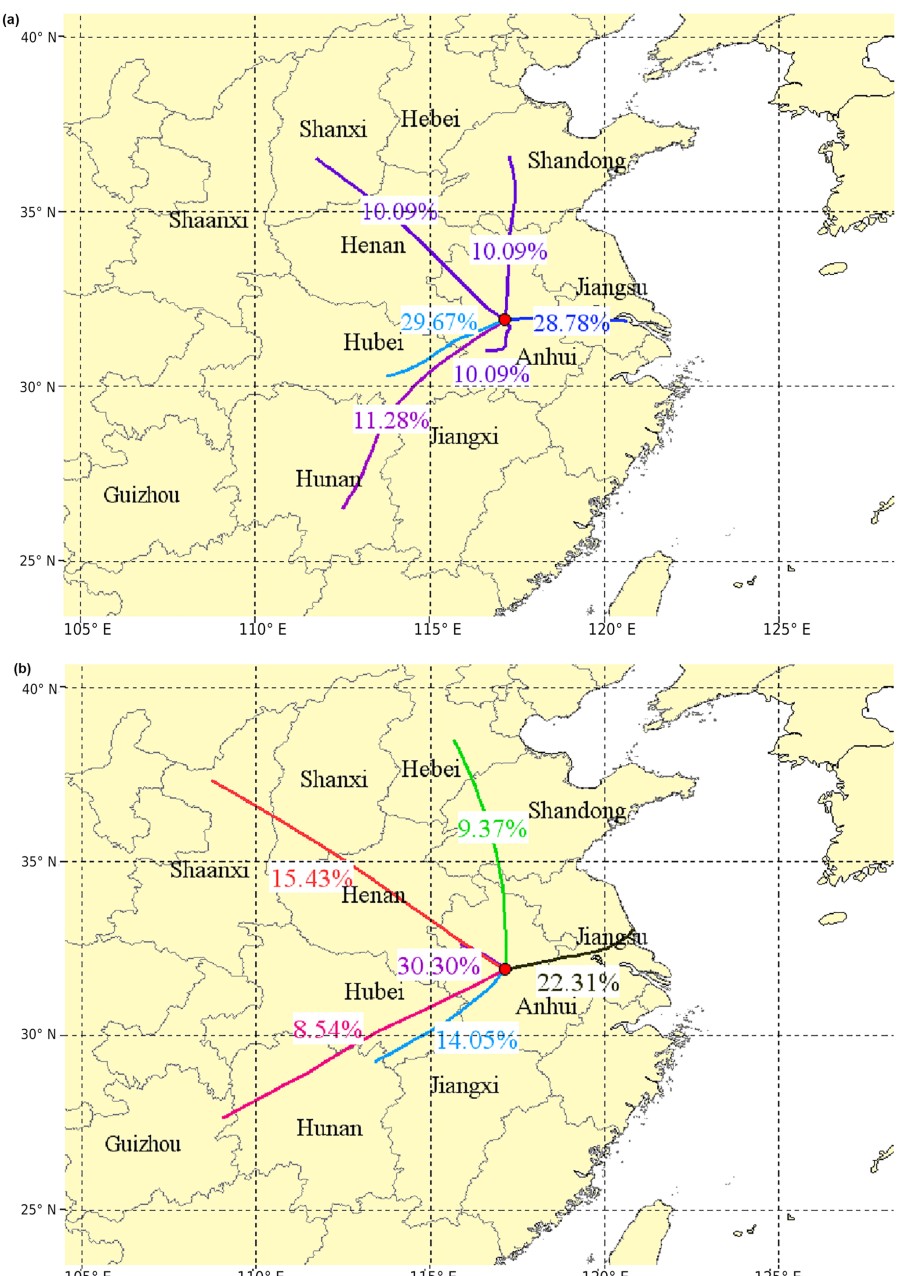

**Figure 2.** One-day HYSPLIT back trajectory clusters arriving at Hefei at 1500 m a.s.l that are coincident with the FTS measurements from 2014 to 2017. **(a)** Spring and summer (MAM/JJA) and **(b)** Autumn and winter (SON/DJF) season. The base map was generated using the TrajStat 1.2.2 software (http://www.meteothinker.com, last access: 23 May 2018).

to the northwest of Anhui province. Trajectories of eastern-origin, western-origin, and northern-origin air masses in SON/DJF are 6.5 %, 13.1 %, and 0.7 % less frequent than the MAM/JJA ones, respectively. As a result, the air masses outside Anhui province have a 20.2 % smaller contribution to the observed tropospheric $O_3$ levels in SON/DJF than in MAM/JJA. In contrast, trajectories of local-origin air masses in SON/DJF are 20.2 % more frequent than the MAM/JJA ones, indicating a more significant contribution of air masses in Anhui province in SON/DJF.

The majority of the Chinese population lives in the eastern part of China, especially in the three most developed regions, the Jing–Jin–Ji (Beijing–Tianjin–Hebei), the Yangtze River Delta (YRD; including Shanghai–Jiangsu–Zhejiang–Anhui), and the Pearl River Delta (PRD; including Guangzhou, Shenzhen, and Hong Kong). These regions consistently have the highest emissions of anthropogenic precursors (Fig. S6),

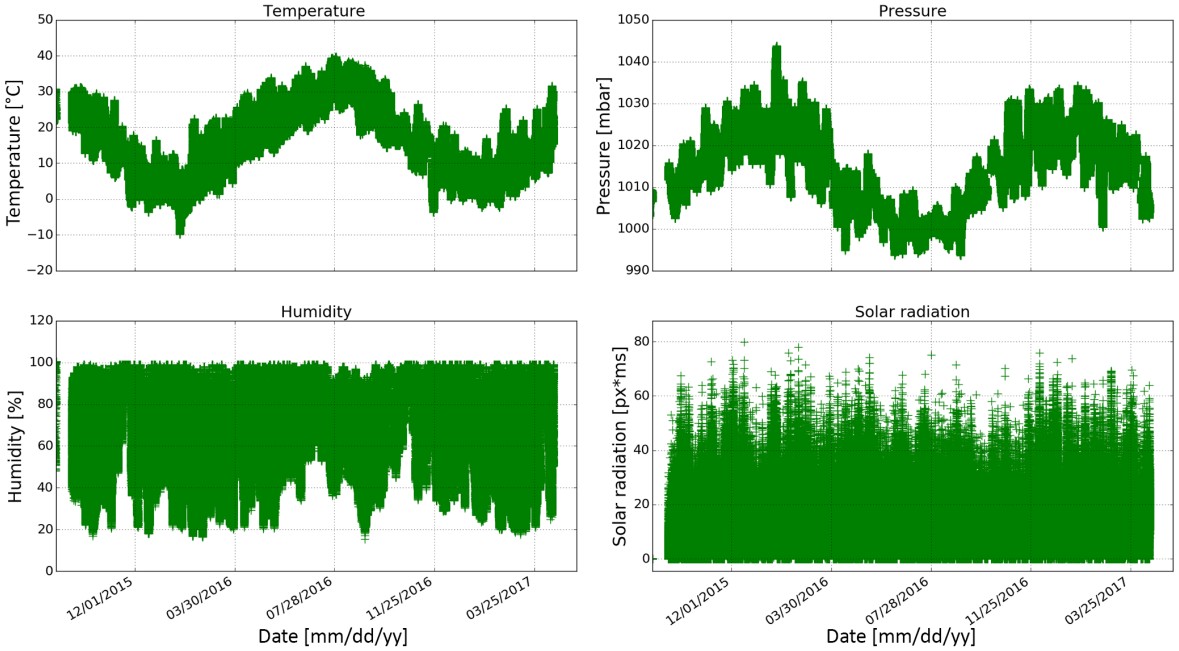

**Figure 3.** Minutely averaged time series of temperature, pressure, humidity, and solar radiation recorded by the surface weather station.

which have led to severe region-wide air pollution. This is particularly the case for the Hefei site, located in the central-western corner of the YRD, where the population in the southeastern area is typically denser than the northwestern area. Specifically, the southeast of Jiangsu province and the south of Anhui province are two of the most developed areas in YRD, and human activities therein are very intense. Therefore, when the air masses originated from these two areas, the $O_3$ level is usually very high. Overall, compared with the SON/DJF season, the more southeastern air masses transportation in MAM/JJA indicated that the observed tropospheric $O_3$ levels could be more influenced by the densely populated and industrialized areas, broadly accounting for the higher $O_3$ level and variability in MAM/JJA.

# 5   Tropospheric $O_3$ production regime

## 5.1   Meteorological dependency

Photochemistry in polluted atmospheres, particularly the formation of $O_3$, depends not only on pollutant emissions but also on meteorological conditions (Lei et al., 2008; Wang et al., 2017; Coates et al., 2016). In order to investigate the meteorological dependency of the $O_3$ production regime in the observed area, we analyzed the correlation of the tropospheric $O_3$ with the coincident surface meteorological data. Figure 3 shows time series of temperature, pressure, humidity, and solar radiation recorded by the surface weather station. The seasonal dependencies of all these coincident meteorological elements show no clear dependencies except for

the temperature and pressure, which show clear reverse seasonal cycles. Generally, the temperatures are higher and the pressures are lower in MAM/JJA than those in SON/DJF. The correlation plots between the FTS tropospheric $O_3$ column and each meteorological element are shown in Fig. 4. The tropospheric $O_3$ column shows positive correlations with solar radiation, temperature, and humidity, and negative correlations with pressure.

High temperature and strong sunlight primarily affects $O_3$ production in Hefei in two ways: by speeding up the rates of many chemical reactions and by increasing emissions of VOCs from biogenic sources (BVOCs) (Sillman and Samson, 1995b). While emissions of anthropogenic VOCs (AVOCs) are generally not dependent on temperature, evaporative emissions of some AVOCs do increase with temperature (Rubin et al., 2006; Coates et al., 2016). Elevated $O_3$ concentration generally occurs on days with wet conditions and low pressure in Hefei, probably because these conditions favor the accumulation of $O_3$ and its precursors. Overall, MAM/JJA meteorological conditions are more favorable to $O_3$ production (higher sun intensity, higher temperature, wetter condition, and lower pressure) than SON/DJF, which supports the fact that tropospheric $O_3$ in MAM/JJA is larger than that in SON/DJF.

## 5.2   PO$_3$ relative to CO, HCHO, and NO$_2$ changes

In order to determine the relationship between tropospheric $O_3$ production and its precursors, the chemical sensitivity of PO$_3$ relative to tropospheric CO, HCHO, and NO$_2$ changes was investigated. Figure 5 shows time series of tropospheric

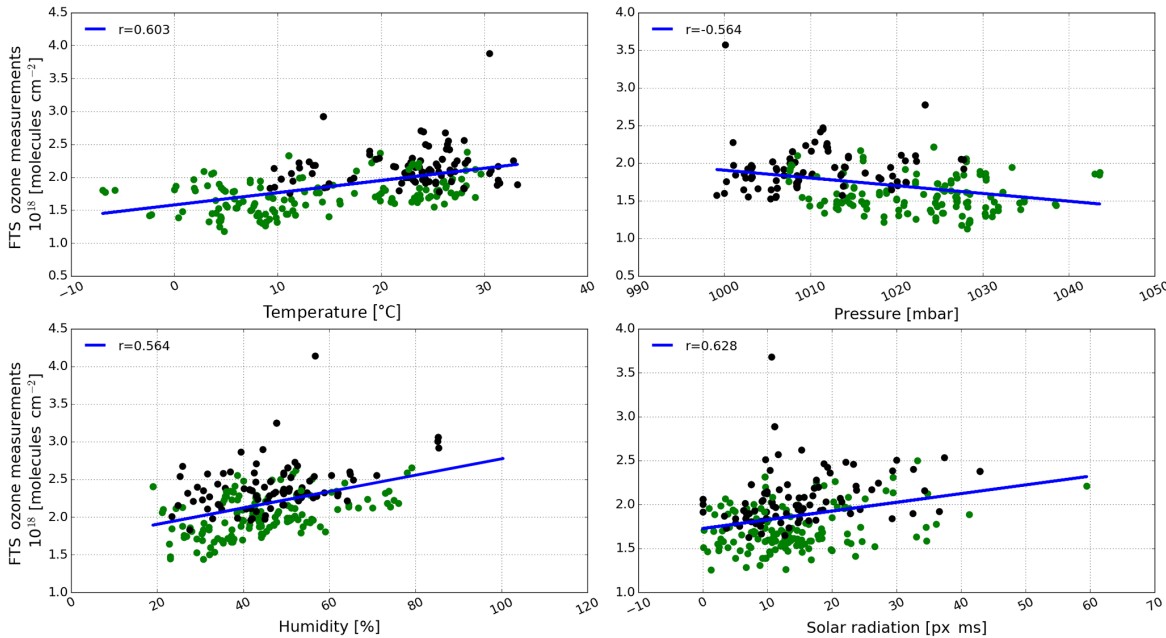

**Figure 4.** Correlation plot between the FTS tropospheric $O_3$ column and the coincident surface meteorological data. Black dots are data pairs within the MAM/JJA season and green dots are data pairs within the SON/DJF season.

CO, HCHO, and $NO_2$ columns that are coincident with $O_3$ counterparts. The tropospheric $NO_2$ was deduced from the OMI product selected within the $\pm0.7°$ latitude/longitude rectangular area around the Hefei site. The retrieval uncertainty for the tropospheric column is less than 30 % (https://disc.gsfc.nasa.gov/datasets/OMNO2_V003/ last access: 23 May 2018). Tropospheric HCHO and $NO_2$ show clear reverse seasonal cycles. Generally, tropospheric HCHO is higher and tropospheric $NO_2$ is lower in MAM/JJA than in SON/DJF. Pronounced tropospheric CO was observed, but the seasonal cycle is not evident, probably because CO emission is not constant over the season or season dependent.

Figure 6 shows the correlation plot between the FTS tropospheric $O_3$ column and the coincident tropospheric CO, HCHO, and $NO_2$ columns. The tropospheric $O_3$ column shows positive correlations with tropospheric CO, HCHO, and $NO_2$ columns. Generally, the higher the tropospheric CO concentration, the higher the tropospheric $O_3$, and both VOCs and $NO_x$ reductions decrease $O_3$ production. As an indicator of regional air pollution, the good correlation between $O_3$ and CO (Fig. 6a) indicates that the enhancement of tropospheric $O_3$ is highly associated with the photochemical reactions which occurred in polluted conditions rather than due to the STE process. The relatively weaker overall correlations of $O_3$ with HCHO (Fig. 6b) and $NO_2$ (Fig. 6c) are partly explained by different lifetimes of these gases, i.e., several hours to 1 day in summer for $NO_2$ and HCHO and several days to weeks for $O_3$. So older $O_3$-enhanced air masses easily loose traces of $NO_2$ or HCHO. Since the sensitivity of $PO_3$ to VOCs and $NO_x$ is different under different

limitation regimes, the relatively flat overall slopes indicate that the $O_3$ pollution in Hefei can be fully attributed neither to $NO_x$ pollution nor to VOC pollution.

## 5.3 $O_3$–$NO_x$–VOC sensitivities

### 5.3.1 Transition/ambiguous range estimation

Referring to previous studies, the chemical sensitivity of $PO_3$ in Hefei was investigated using the column HCHO/$NO_2$ ratio (Martin et al., 2004; Duncan et al., 2010; Witte et al., 2011; Choi et al., 2012; Jin and Holloway, 2015; Mahajan et al., 2015; Schroeder et al., 2017; Jin et al., 2017). The methods have been adapted to the particular conditions in Hefei. In particular the findings of Schroeder et al. (2017) have been taken into account.

Since the measurement tools for $O_3$ and HCHO, the pollution characteristic, and the meteorological condition in this study were not the same as those of previous studies, the transition thresholds estimated in previous studies were not applied here (Martin et al., 2004a; Duncan et al., 2010; Witte et al., 2011; Choi et al., 2012; Jin and Holloway, 2015; Mahajan et al., 2015; Schroeder et al., 2017; Jin et al., 2017). In order to determine transition thresholds applicable in Hefei, China, we iteratively altered the column HCHO/$NO_2$ ratio threshold and judged whether the sensitivities of tropospheric $O_3$ to HCHO or $NO_2$ changed abruptly. For example, in order to estimate the VOC-limited threshold, we first fitted tropospheric $O_3$ to HCHO that lies within column HCHO/$NO_2$ ratios < 2 (an empirical starting point) to obtain the corre-

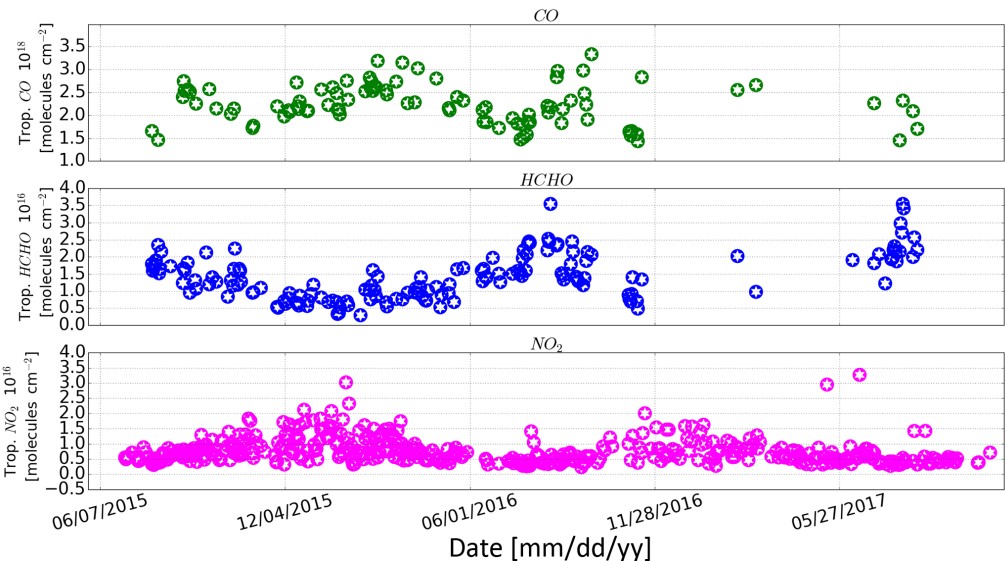

**Figure 5.** Time series of tropospheric CO, HCHO, and $NO_2$. Tropospheric CO and HCHO were derived from FTS observations, which is the same as tropospheric $O_3$, and tropospheric $NO_2$ is derived from OMI data.

sponding slope and then we decreased the threshold by 0.1 (an empirical step size) and repeated the fit, i.e., only fitted the data pairs with column $HCHO/NO_2$ ratios < 1.9. This was done iteratively. Finally, we sorted out the transition ratio which shows an abrupt change in slope, and regarded this as the VOC-limited threshold. Similarly, the $NO_x$-limited threshold was determined by iteratively increasing the column $HCHO/NO_2$ ratio threshold until the sensitivity of tropospheric $O_3$ to $NO_2$ changed abruptly.

The transition threshold estimation with this scheme exploits the fact that $O_3$ production is more sensitive to VOCs if it is VOC-limited and is more sensitive to $NO_x$ if it is $NO_x$ limited, and there exists a transition point near the threshold (Martin et al., 2004a). Su et al. (2017) used this scheme to investigate the $O_3$–$NO_x$–VOC sensitivities during the 2016 G20 conference in Hangzhou, China, and argued that this diagnosis of PO$_3$ could reflect the overall $O_3$ production conditions.

### 5.3.2   PO$_3$ limitations in Hefei

Through the above empirical iterative calculation, we observed a VOC-limited regime with column $HCHO/NO_2$ ratios < 1.3, an $NO_x$-limited regime with column $HCHO/NO_2$ ratios > 2.8, and a mixed VOC–$NO_x$-limited regime with column $HCHO/NO_2$ ratios between 1.3 and 2.8. Column measurements sample a larger portion of the atmosphere, and thus their spatial coverage is larger than in situ measurements. So the photochemical scene disclosed by column measurements is larger than the in situ measurement. Specifically, this study reflects the mean photochemical condition of the troposphere.

Schroeder et al. (2017) argued that the column measurements from space have to be used with care because of the high uncertainty and the inhomogeneity of the satellite measurements. This has been mitigated in this study by the following.

The FTS measurements have a much smaller footprint than the satellite measurements. Also, we concentrate on measurements recorded during midday, when the mixing layer has largely been dissolved.

The measurements are more sensitive to the lower parts of the troposphere, which can be inferred from the normalized averaging kernels (AVKs). The reason is simply that the AVKs show the sensitivity to the column, but the column per altitude decreases with altitude.

Figure 7 shows time series of column $HCHO/NO_2$ ratios which varied over a wide range from 1.0 to 9.0. The column $HCHO/NO_2$ ratios in summer are typically larger than those in winter, indicating that the PO$_3$ is mainly $NO_x$ limited in summer and mainly VOC limited or mixed VOC–$NO_x$ limited in winter. Based on the calculated transition criteria, 106 days of observations that have coincident $O_3$, HCHO, and $NO_2$ counterparts in the reported period are classified, where 57 days (53.8 %) are in the MAM/JJA season and 49 days (46.2 %) are in the SON/DJF season. Table 4 lists the statistics for the 106 days of observations, which shows that $NO_x$-limited, mixed VOC–$NO_x$-limited, and VOC-limited PO$_3$ accounts for 60.3 % (64 days), 28.3 % (30 days), and 11.4 % (12 days), respectively. The majority of $NO_x$-limited (70.3 %) PO$_3$ lies in the MAM/JJA season, while the majority of mixed VOC–$NO_x$-limited (70 %) and VOC-limited (75 %) PO$_3$ lies in the SON/DJF season. As a result, reductions in $NO_x$ and VOC could be more effective to mitigate

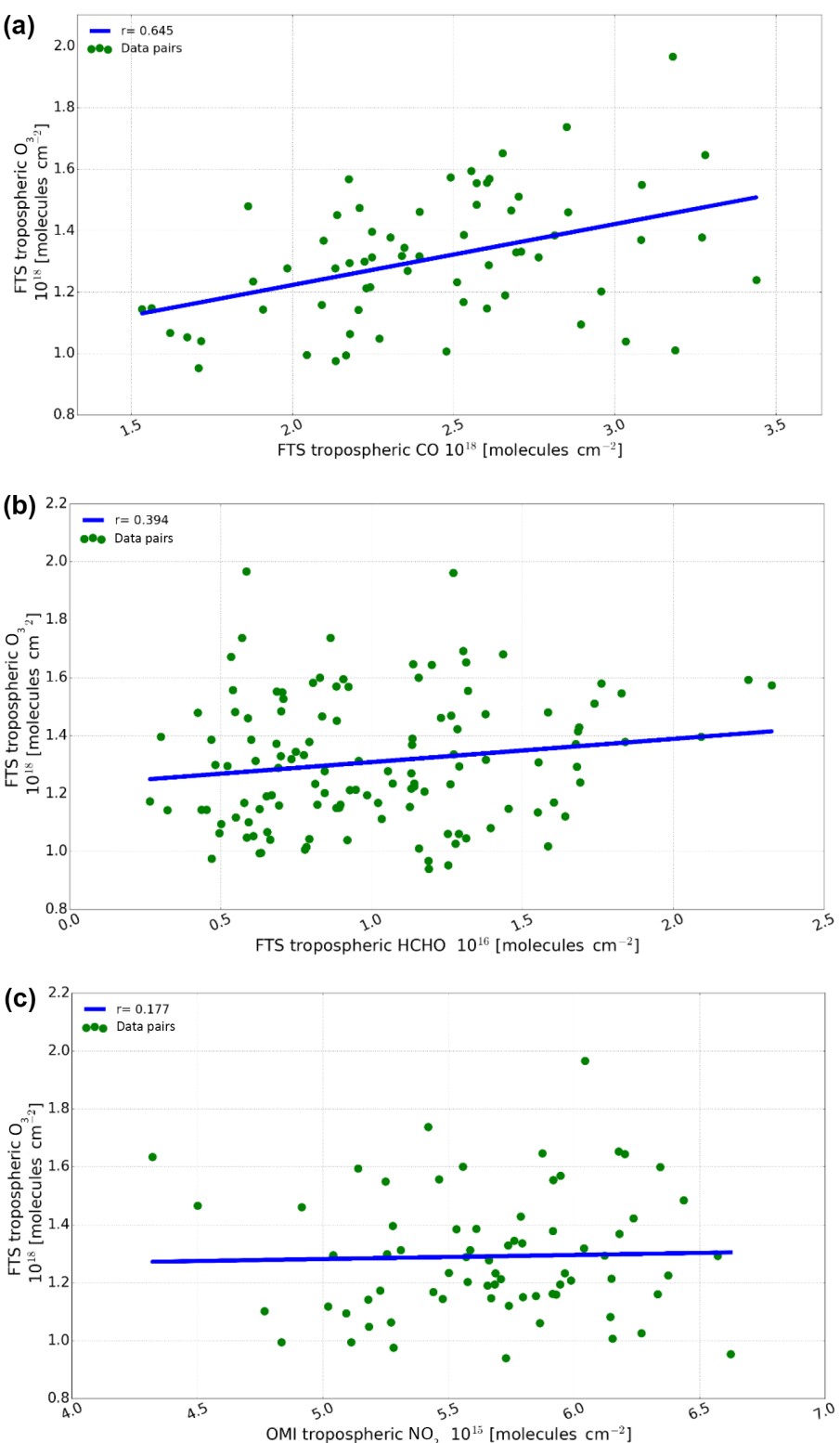

**Figure 6.** Correlation plot between the FTS tropospheric $O_3$ column and coincident tropospheric CO (**a**), HCHO (**b**), and $NO_2$ (**c**) columns. The CO and HCHO data are retrieved from FTS observations, and the $NO_2$ data were deduced from the OMI product.

**Table 4.** Chemical sensitivities of PO₃ for the selected 106 days of observations that have coincident O₃, HCHO, and NO₂ counterparts.

| Items | Proportion | | Autumn and winter | | Spring and summer | |
|---|---|---|---|---|---|---|
| | days | percentage | days | percentage | days | percentage |
| $NO_x$ limited | 64 | 60.3 % | 19 | 29.7 % | 45 | 70.3 % |
| Mixed VOC–$NO_x$ limited | 30 | 28.3 % | 21 | 70 % | 9 | 30 % |
| VOC limited | 12 | 11.4 % | 9 | 75 % | 3 | 25 % |
| Sum | 106 | 100 % | 49 | 46.2 % | 57 | 53.8 % |

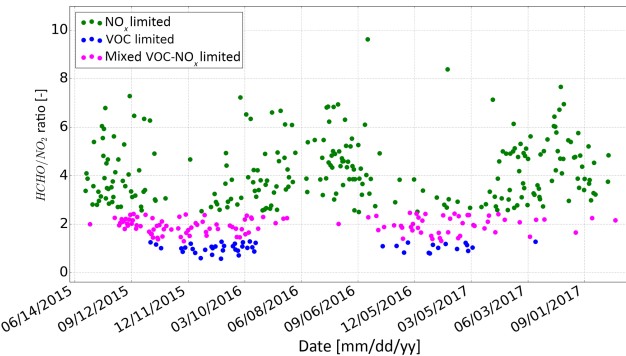

**Figure 7.** Time series of column HCHO/NO₂ ratios.

O₃ pollution in the MAM/JJA and SON/DJF seasons, respectively. Furthermore, considering most of PO₃ is $NO_x$ limited or mixed VOC–$NO_x$ limited, reductions in $NO_x$ would reduce O₃ pollution in eastern China.

## 6 Conclusions

We investigated the seasonal evolution and photochemical production regime of tropospheric O₃ in eastern China from 2014 to 2017 by using tropospheric O₃, CO, and HCHO columns derived from Fourier transform infrared spectrometry (FTS), the tropospheric NO₂ column deduced from the Ozone Monitoring Instrument (OMI), the surface meteorological data, and a back trajectory cluster analysis technique. A pronounced seasonal cycle for tropospheric O₃ is captured by the FTS, which roughly increases over time in the first half year and reaches the maximum in June, and then it decreases over time in the second half year. Tropospheric O₃ columns in June are $1.55 \times 10^{18}$ molecules cm$^{-2}$ (56 DU (Dobson units)), and in December they are $1.05 \times 10^{18}$ molecules cm$^{-2}$ (39 DU). Tropospheric O₃ columns in June were ∼ 50 % higher than those in December. A broad maximum within both spring and summer (MAM/JJA) is observed, and the day-to-day variations in MAM/JJA are generally larger than those in autumn and winter (SON/DJF). This differs from tropospheric O₃ measurements in Vigouroux et al. (2015). However, Vigouroux et al. (2015) used measurements at relatively clean sites.

Back trajectory analysis showed that air pollution in Jiangsu and Anhui provinces in eastern China; Hebei and Shandong provinces in northern China; Shaanxi, Henan, and Shanxi provinces in northwest China; and Hunan and Hubei provinces in central China contributed to the observed tropospheric O₃ levels. Compared with the SON/DJF season, the observed tropospheric O₃ levels in MAM/JJA are more influenced by the transport of air masses from densely populated and industrialized areas, and the high O₃ level and variability in MAM/JJA is determined by the photochemical O₃ production. The tropospheric-column HCHO/NO₂ ratio is used as a proxy to investigate the chemical sensitivity of the O₃ production rate (PO₃). The results show that PO₃ is mainly nitrogen oxide ($NO_x$) limited in MAM/JJA, while it is mainly VOC or mixed VOC–$NO_x$ limited in SON/DJF. Reductions in $NO_x$ and VOC could be more effective to mitigate O₃ pollution in the MAM/JJA and SON/DJF seasons, respectively. Considering most of PO₃ is $NO_x$ limited or mixed VOC–$NO_x$ limited, reductions in $NO_x$ would reduce O₃ pollution in eastern China.

*Data availability.* The SFIT4 software can be found via https://www2.acom.ucar.edu/irwg/links (last access: 23 May 2018). The data used in this paper are available on request.

**The Supplement related to this article is available online at https://doi.org/10.5194/acp-18-1-2018-supplement.**

*Author contributions.* The first two authors contributed equally to this work. YS and CL prepared the paper with inputs from all coauthors. MP, CV, JN, NJ, and MDM designed the retrieval and optimized the content. QH and WS conceived ozone production regime study. WZ provided the OMI NO₂ product. WW, CS, YT, XX, MZ, and JL carried out the experiments and performed back trajectory cluster analysis.

*Competing interests.* The authors declare that they have no conflict of interest.

*Special issue statement.* This article is part of the special issue "Quadrennial Ozone Symposium 2016 – Status and trends of atmospheric ozone (ACP/AMT inter-journal SI)". It is a result of the Quadrennial Ozone Symposium 2016, Edinburgh, United Kingdom, 4–9 September 2016.

*Acknowledgements.* This work is jointly supported by the National High Technology Research and Development Program of China (no. 2016YFC0200800, no.2018YFC0213104, no. 2017YFC0210002, no. 2016YFC0203302), the National Science Foundation of China (no. 41605018, no.41877309, no. 41405134, no.41775025, no. 41575021, no. 51778596, no. 91544212, no. 41722501, no. 51778596), the Anhui Province Natural Science Foundation of China (no. 1608085MD79), the Outstanding Youth Science Foundation (no. 41722501), and the German Federal Ministry of Education and Research (BMBF) (grant no. 01LG1214A). The processing and post-processing environments for SFIT4 are provided by the National Center for Atmospheric Research (NCAR), Boulder, Colorado, USA. The NDACC networks are acknowledged for supplying the SFIT software and advice. The HCHO micro-windows were obtained at BIRA-IASB during the ESA PRODEX project TROVA (2016–2018) funded by the Belgian Science Policy Office. The LINEFIT code is provided by Frank Hase, Karlsruhe Institute of Technology (KIT), Institute for Meteorology and Climate Research (IMK-ASF), Germany. The authors acknowledge the NOAA Air Resources Laboratory (ARL) for making the HYSPLIT transport and dispersion model available on the Internet. The authors would also like to thank Jason R. Schroeder and three anonymous referees for useful comments that improved the quality of this paper.

Edited by: Stefan Reis
Reviewed by: five anonymous referees

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

**Remarks from the language copy-editor**

CE1   Thank you for your clarification. I have made changes following your explanation, but please double-check these as I am not quite sure I have understood them correctly ($\mathbf{S}_a$, for example, is repeated again in the next sentence – is it correct here?).

**Remarks from the typesetter**

TS1   The changes to the table need to be approved by the handling editor.