# Peer review of "Ozone seasonal evolution and photochemical production"

_Atmospheric Chemistry and Physics, 2017_

## Short Comment (SC1) · 20 Dec 2017

The authors may wish to refer to recent work by Schroeder et al. (2017) doi: 10.1002/2017JD026781 , who did a thorough analysis of the utility of column HCHO/NO2 ratios to indicate surface O3. Schroeder et al. found that column HCHO/NO2 ratios are not effective for determining surface O3 sensitivity, as the "transition" range is certainly larger than the range of 1-2 described in Duncan et al (2010), and larger than the 1.3-2.8 used in this work.

---

## Referee Comment (RC1) · Anonymous Referee #1 · 5 Feb 2018

General comments :

The authors have used a new FTIR dataset to infer tropospheric ozone seasonal evolution and photochemical production regime at Hefei in China. Comparisons of the new dataset with OMI observations, and the GEOS-Chem and WRF-chem model data have shown good agreements. Back trajectories analyses have been used to attribute the contribution regions, and seasonal variabilities, to the high ozone levels observed at Hefei. The chemical sensitivity to ozone production has been studied at Hefei by using proxies such as CO and HCOH.

[Figure]

Although the authors use a new dataset, the novelty of some results is hard to admit. For instance, it is presented the fact that tropospheric ozone column is higher in spring/summer as a key result, which is a known scientific idea (same for better agreements comparing smoothed profiles relative to unsmoothed profiles). A reorganization of the paper's structure is needed, with less focus on the know results and more thinking about what is the paper contribution to scientific progress. In addition, the goal of the comparisons of the new dataset with independent data (atmospheric models and satellite observations) is unclear, as well as the use of two different model (global and regional). The objectives of the paper should be clarified and listed in a concise way. The number of figures should be reduced to fit the main scientific results. Discussions about results, such as model and observations comparisons, are missing and would improve the scientific impact of the paper.

The thorough section describing the retrievals is well written and I would advise the authors to submit this paper to a more technical journal, such as Atmospheric Measurement Technique, if not addressing these comments.

Specific comments:

Concerning the structure of the paper, it needs to be reorganized with a shorter abstract focusing a key results, more detailed introduction about the proxies used to assess the chemical sensitivity to ozone production, more sub-sections and tables summarizing the results of the comparisons, less figures, and appropriate English language. Consistency is also needed across the paper: define acronyms once (NDACC has never been defined in the abstract but appears in the keyword section, same for HCOH and VOC in the abstract, . . .) and use them along the manuscript (ozone or O3 ?). Change old references with newest and avoid Wikipedia as a reference.

In the introduction section, more explanations about why using proxies such as CO and HCOH would help the reader understanding how to assess the chemical sensitivity to ozone production. In the retrieval strategy section, you mention a meteorological

station onsite. Do you correct the NCEP profiles with these data? If yes, it should be clarified.

Define the use of the coincidence criteria when comparing to OMI (Why 3 hours and 0.7°?) and other independent data.

In section 5.1, it is mentioned a trend analysis on a 4-years timeseries. The world "trend" needs to be changed.

Are back trajectories used to investigate the regions of influence of high tropospheric ozone at Hefei? If yes, it needs to be clarified and better structured in a sub-section. The end of section 5.1 needs to be better structured to emphasis on the scientific conclusions.

Concerning the comparisons with models, you may want to clarify their use; it is unclear if it is to assess the new dataset quality or investigate the model performances to reproduce observations. Explain the scientific interest of comparing the FTIR dataset with a global and a regional model.

Discussions about results concerning comparisons between model/satellite and FTIR observations are missing and would raise the scientific level of the paper. Why is there a shift in the seasonal maximum in GEOS-Chem? Why do OMI and the FTS exhibit different seasonality? For WRF-Chem it is mentioned that the difference could be attributed to uncertainties in the input files, but what about the meteorological data, and/or the chemistry? This has to be further analyzed and explained.

Technical corrections :

- line 20 and 22 : define acronym CO, HCOH, and NO2

- line 26 : by "the" FTS

- line27 : "occur" with no s

- line 32 and 33 : choose the precision, one or two decimal?

- line 34 : by "atmospheric models" GEOS-Chem and WRF-Chem

- line 41-43 : rephrase the sentence

- line 45-50 : state that HCOH is a VOC and define VOC

- key words : NDACC never defined in the abstract

- line 55 : add a reference

- line 56 : add a reference

- line 53-71 : references are old

- line 75-77 : why so many references? Are they all relevant? You may use the most relevant one

- line 78-79 : define all chemical species

- line 84 : state the accuracy or use a reference

- line 87 : avoid Wikipedia as a scientific reference

- line 88 : first time ozone is written O3. Be consistent across the manuscript

- line 91 : "PM2.5"

- line 93 : "the" FTS

- line 96 : "Most NDACC sites"

- line 99 : Is the Hefei site a NDACC site? It is not clear here

- line 103 : add a reference for OMI

- line 105 : "the" site description

- line 107 : clarify the sentence "ozone related gases"

- line 114 : reference to Figure 1b

- line 114-115 : rephrase

- line 117 : clarify why it is an important region

- line 118 : add a reference

- line 119-120 : rephrase

- line 121 : change "the same as" to "similar to"

- line 123-125 : rephrase

- line 125 : change "demonstrated" to "showed"

- line 126 : "typical observation day in August"

- line 139 : define MIR

- line 143 : "for O3 measurements"

- line 144 : are you certain filters are used to avoid detector non-linearity? What about signal to noise ratios?

- line 148 : clarify "ozone related gases"

- line 151 : how much is an adequate accuracy?

- line 152-155 : is this sentence misplaced ? If not explain why it "confirms"

- line 155 : delete "so"

- line 157 : delete "overpass"

- line 161 : add a reference to SFIT4

- line 195 : use mathematical equations

- line 230 : "Figures 4 and 5"

- line 252 : explain why two sets of models

- line 282 : how much is 0.7° in kilometers at Hefei?

- line 307 : add the GEOS-Chem general reference

- line 320 : what is the nearest grid in kilometer?

- line 345 : add a general reference for WRF-Chem

- line 347 : "Liu et al. (2016)"

- line 348 : "20 x 20 km"

- line 382-383 : add a reference

- line 395 : delete "global"

- line 430 : "air pollution"

- line 430-432 : rephrase the sentence

- line 435 : state the percentage

- line 439 : "considering the fact"

- line 441 : "air pollution"

- line 454 : "Obvious". Why it is obvious?

- line 465 : "not an emission pollutant" is not clear, rephrase

- line 466-467 : explain why the fact that it is complicated means that it shows regional representativeness?

- line 479 : "as a result", explain further the link between the two sentences

- line 485 : stay at present

- line 497 : how much are a good and an adequate correlation?

- line 502 : "has taken"

- line 505-513 : this could go to the introduction section

- line 525 : change "obtainment"

- line 554 : change "validate" since OMI, GEOS-Chem, and WRF-Chem, to my knowledge, have already been validated

- Figure 1 a : change to see star colors

- Figure 1 b : instead of showing SZA for 1 day, you may want to show the mean SZA for all the data involved in this study

- Figure 1 legend : is it relevant to point out the wetlands? Are the red hexagons SZA or azimuth angle?

- Figure 2 : cut altitude at 60 or 80 km

- Figure 3 : arrange the figure so that the text is readable and is not crossed by the lines. Figure 3 partial column averaging kernel of HCOH : explain what are the influences on retrieved column of a partial Avk of 12

- Figure 4 and 5 : cut at 60 or 80 km and combine them in one Figure

- Figure 6, 8, and 10 : insert the number of points included in the comparison and insert the standard deviation of the mean

- Figure 7 and 9 a : insert error bars

- Figure 7 b : shift text

- Figure 11 b : why showing both biased and unbiased data?

- Figure 12 b : reduce y-axis scale

- Figure 15 : maybe plot all the measurements involved instead of daily means?

- Figure 16 : reduce the size of the dots. Do you consider error bars to fit the data?

[Figure]

- Figure 18 : all 3 panels should fit in one page. Do you account for uncertainties when fitting the data? R = 919 with 8 points, are you certain it is a robust statistic?

- Table 1 : enlarge the first column to adjust the word "regularization". O3668 with exponent and index
* * *

---

## Referee Comment (RC2) · Anonymous Referee #2 · 5 Feb 2018

**Review of "Ozone seasonal evolution and photochemical production regime in polluted troposphere in eastern China derived from high resolution FTS observations", Sun et al., ACPD.**

**Summary:**

The authors report on solar absorption FTIR measurements of tropospheric columns of O3, CO, and HCHO at a candidate NDACC IRWG observation station in Hefei, China. High spectral resolution measurements were conducted between 2014 and 2017 and fill a data gap within the NDACC observation network. The data shows higher tropospheric O3, also with higher variability, in spring and summer. The authors compare these O3 measurements to OMI satellite O3 (PROFOZ product), as well as GEOS-Chem (2 x 2.5 deg) and WRF-Chem (20 x 20 km) model O3 outputs. Comparisons are done in both profile and tropospheric partial column form.

Ozone FTS vs. GEOS-Chem model differences (481 coincidences) are attributed to uncertainties in GEOS-Chem input files ("ozone production loss rates and emission inventory"), it is concluded that GEOS-Chem is biased 13% lower (along profile), with r=0.5 for tropospheric column correlation plots.

Ozone FTS vs. WRF-Chem model differences (481 coincidences) are attributed to uncertainties in WRF-Chem input files ("ozone production and loss rates and MEIC inventory"), it is concluded that WRF-Chem is biased 12% lower (along profile), with r = 0.65 for tropospheric column correlation plots.

Comparisons to coincident OMI ozone profiles and partial (tropospheric) OMI columns were done on 53 coincident measurements after filtering for 0.7° spatial coincidence. Coincident FTS profiles were averaged in a 3 hour window around the OMI overpass at 13:30. OMI profiles were smoothed with FTIR averaging kernels. The OMI profiles are biased 2-13% lower than FTIR profiles, with r=0.73 for tropospheric column correlation plots, in which most OMI points sit below the 1:1 line, indicating also a low bias of OMI w.r.t. FTS.

Both sets of model ozone data are described as "smoother" than FTIR data and are "bias corrected" by adding a constant offset to the tropospheric O3 columns throughout the year to shift the model data towards FTIR partial column values. GEOS-Chem partial columns are increased by ~100% while WRF-Chem partial columns are increase by ~33% to increase agreement with FTS. Finally, OMI ozone partial column data were increased by ~20% and only then were monthly mean ozone partial column differences calculated.

24-hour back trajectories were calculated arriving at Hefei at 3000 m.a.s.l. from 2014-2017, presumably for those days with FTS observations (?), and they were grouped into spring/summer (presumably MAM/JJA?) and autumn/winter (presumably SON/DJF?). Summer transport is less vigorous and more varied than winter transport, as expected, bringing more air from highly polluted areas, e.g., east China, and broadly accounting for higher O3 and higher O3 variability in the data in spring/summer.

Finally, the O3 production regime is analyzed by describing correlations to meteorological variables recorded at Hefei, as well as looking at O3 vs. CO, O3 vs NO2 (for ratios of HCHO/NO2 > 2.8, assumed to correspond to NOx-ltd O3 production) and O3 vs HCHO (for ratios of HCHO/NO2 < 1.3, assumed to correspond to VOC-ltd O3 production). The ratios to indicate the O3 production regime were found iteratively until the correlation between O3 and NO2 or O3 and HCHO was > 0.6. 106 days of observations (O3, HCHO, CO from FTS; NO2 from OMI) were identified and of those 60% were NOx-ltd, 11% were VOC-ltd, and the remainder were mixed.

**Major comments:**

The paper is generally well written and presents a thorough error budget and sensitivity analysis of FTIR retrievals (O3, CO, HCHO) from a new candidate station in the NDACC network. The methods used here are well known and figures 2-5 should also move to the appendix, along with the Rodgers & Connor formulation, unless the authors highlight how their averaging kernels and error budget profiles differ from other similar published results. The paper presents a valuable new and growing observational dataset, however, this reviewer recommends major revisions in order to meet the ACP criteria of scientific significance and quality.

The FTS O3 measurements are higher than both models (global and regional) and the OMI measurements. The FTS measures a total column through a particular atmospheric slant column, and is expected to be less sensitive to local O3 events than an in situ sensor. We expect generally good agreement with downward-looking OMI, although coincidences are always a challenge. We also expect differences in the FTS vs. model comparisons because of different representativeness offered by a 20x20 km model vs. as 2.5° x 2° model. This is not discussed in the paper. Also, for the 20x20 km WRF-Chem model, the profile up to 10 km could extend over two horizontal grid boxes for most SZAs > 45°, depending on the location of Hefei within a model grid box. Has this been considered?

Without discussing representativeness, the authors attribute FTS vs. model differences to model "input files", e.g., "ozone production loss rates and emission inventory" which is superficial. As a consequence, we learn little, if anything, about specific model processes and emission inventories that may be responsible. Also, why is the data from this candidate station considered as "truth" in the comparison to OMI and the models? The total errors are estimated as 10% but they are dominated by smoothing error and based on very tight Sa values for O3 (10%), so (as the authors note), they are an underestimate. If the authors plotted OMI vs. FTS trop O3 column data with both data sets' error bars they would still not overlap, but presumably OMI data has been validated – is it generally found to be low compared to other data?

The addition of a simple offset to model O3 values before looking at fractional monthly mean differences w.r.t. FTS is problematic because it is evident in figures 9 and 11 that such a simple manipulation does not bring the data points onto the 1:1 line. Instead, we have the highest O3 values below the FTS measurements and the lowest values above. This is even more dramatic in GEOS-Chem data, presumably because of lower model resolution, which homogenizes high O3 values over a large grid cell, while raising the background O3 values. Since the highest values occur in spring/summer and the lowest in autumn/winter, the bias is seasonally dependent and therefore not just due to spatial representativeness. Is it due to incorrect emissions or chemistry? What are the main chemistry and emissions differences between the two models being compared to FTS? WRF-Chem is running with the MEIC inventory, presumably optimized for China, as well as biogenic emissions from MEGAN – why does it only do a little bit better than GEOS-Chem?

About smoothing the OMI profile by the FTIR averaging kernels, this method is meant to be applied to high vertical resolution correlative data, which OMI is not. It has about ~1 DOF in the troposphere itself. This may explain why there is still a lot of "shape" left in the fractional difference between FTIR and smoothed OMI profiles. What do OMI kernels look like and where is its peak of sensitivity – is it the same as for FTS?

The trajectory cluster analysis is difficult to follow without familiarity with China's geography. That can easily be fixed by adding the major city and region names referred to in the discussion to Figure 13. Without this information, it is hard to quickly judge if 1-day trajectories are long enough for transport to occur to Hefei. It is also not clear how the trajectories are clustered and the mean cluster trajectories (in color) are hard to see. Another way to represent this data would be to count trajectory elements crossing, e.g., 0.5° x 0.5° grid boxes. Also, why 3000 m? That seems much higher than the typical boundary layer height in winter, and probably also in summer. This choice will influence strongly both the speed and footprint of the pollution regions influencing Hefei. Have the authors tried 1500 m?

Finally, regarding O3 production regimes, ratios of HCHO/NO2 were varied until the correlation was > 0.6 in plots of O3 vs. HCHO and O3 vs. NO2. The outcome is that the correlation for the NOx-ltd plot of O3 vs. NO2 is 0.66 (moderate) while the correlation for the VOC-ltd plot of O3 vs. HCHO is 0.92, with far fewer points remaining in the fit. This seems rather arbitrary and needs justification. Also of the 106 days available for this analysis, which are from spring/summer and which are from autumn/winter? Are all VOC-ltd days in winter?

When I look at the full O3 data in Figure 12, I wonder why there isn't a stronger signature of JJA O3 enhancements in Hefei? (Is it related to filtering out days affected by haze, App B?) Many high values seem to be in May, although the x-axis is hard to read and should really be changed to, Jan 1, June 1, etc., throughout the paper where dates are shown. Or possibly at boundaries between MAM, JJA, SON, DFJ, if these are the groupings for the seasons in the paper. Have the FTS partial columns been compared to in situ O3 monitors in Hefei to see if they also show enhancements in May/June 2015 and 2016? What about the low values in Jan 2015 and 2017 vs. the higher ozone in Jan 2016?

Finally, the Pearson coefficient of 0.35 – 0.6 was taken to mean "moderately correlated" in this work. Typically moderate correlation is associated with values of 0.5 – 0.8, since the lower bound would mean that the model fit to the data explains only 25% of the variations in the data. At 0.35 that drops to only 12%.

**Further detailed technical comments:**

Fig. 1a: most names in this figure are illegible. Use a cleaner map to reduce clutter.

Fig. 1b: no red hexagons are visible, but I assume the red arc is the azimuth and the un-described yellow circles are the SZA.

Fig. 2: what does "with measured ILS" mean in this caption? Is the ILS characterized with linefit and then fixed in the retrievals, or are some ILS parameters still being retrieved? Why is there a loss of sensitivity to HCHO right at the surface? Is this a priori related?

Fig. 3: the HCHO trop column AK seems unhealthy for growing so far past 1 quickly above ~3 km, even if there is little HCHO there. What is the reason for this shape?

Fig. 4: What is the explanation for the peak in the CO error at around ~3 km?

Fig. 5: Legend seems reversed for total random error and z shift for CO.

Fig. 9 and 11: it's hard to judge seasons with the date labels as presented. Also, why do these figures not have the identical number of O3 data points if they are derived from the same data filtering applied to FTS data that is described in App B?

Fig. 14: is based on Fig 12, not 13 as the caption says. Again, what are the model process and inventory differences leading to this? Panel a) says smoothed model, but is OMI not also smoothed in this figure?

Fig. 15: The wind sensor appears to be installed in a poor location as the wind speed never exceeds 0.3 m/s or ~1 km/h! If that is the case, then the wind direction data is also spurious. That's too bad, because I wanted to see a plot of Hefei O3 vs. wind direction to see if O3 is higher when winds blow from the city.

Fig 16: In spite of problems above, the highest O3 values occur for the lowest of the low wind speeds, pointing to the accumulation of local pollution. There is a "moderate" negative correlation between O3 and RH – why? We could learn more if these data were colored according to spring/summer and autumn/winter.

Fig. 19: hard to judge seasons with x-axis labels. Panel b is based on data in panel a that does not seem to sample seasons evenly. This should be discussed.

Table 1: retrieved interfering gases → as columns, I presume, except for H2O, as noted? Also, WM → MW. I'm not sure what footnote b means, please clarify.

**Manuscript:**

P1L74: sun spectra → solar absorption spectra

P1L3: replace wiki reference with something from the many, many refereed papers on Chinese modernization and growing air pollution problems.

P4L89: what are China's AQ standards in ppb for long- and short-term exposure?

P4L95: greatly contribute to ozone pollution controls → contribute to the evaluation of O3 pollution controls

P4L117: … after it is itself validated as an NDACC site and it moves from candidate to regular status.

P5L129: then increases → then SZA increases

P5L129-133: what region influences the measurements depends on the azimuth of observation, yes, but also on the direction and wind speed pushing air masses above Hefei, especially for the lowest parts of the atmosphere. This could be significant when local pollution events are occurring as some events can be completely swept away from the FTS obs path.

P6L173: cited references missing from references section

P6L178: please explain deweighting more clearly. What are instrument SNR levels without deweighting?

P7L187: how are the Sa diagonal element magnitudes chosen? WACCM?

P7L191: is the ILS retrieved in all retrievals or is it done with LINEFIT and then held constant?

P11L315-317: tagged O3 runs are mentioned, which would be nice and would allow the attribution of pollution to various source regions, but these 3 lines are very unclear (i.e., also about restart files)

P14L393-4: basically reproduced … but with slight shifts in timing → July is wrong in both models; why are they low in August? When is the local Hefei smog season?

P14L403: Logan (1985) "observed" → I presume this is a model study?

P16L448: basically consistent throughout all seasons → it really doesn't look like that to me; would be easier to think about if time series started with MAM as opposed to JFM.

P16L457-63: this really is a shallow explanation of what may be causing the differences, from which we learn nothing concrete. Also, how does larger air pollution increase uncertainty in either emission inventories or the photochemical regime? Isn't the latter, especially, something that is diagnosed from the emission rates and relative abundances of NOx and VOCs?

P17L495/6: which emissions are being discussed here: biogenic? anthropogenic? What are the expected magnitudes and timing of each?

P18L528: straightly applied → straight forwardly applied

P19L554: "validate OMI" → that's a strong statement given the unproven nature of these particular FTIR measurements, and given there's no reference to other OMI validation efforts and what they have typically revealed

P19L560: WRF-Chem agreement is "better" → it has a lower "bias" but greater summer differences. It's not clear if that is better given it is a high res model using optimized emissions for China

P22 L651: would not screening out hazy days eliminate a lot of JJA O3 pollution days? Haze isn't a problem for FTIR as much as non-constant intensity (e.g., clouds floating by during a ~20 minute observation time).

---

## Referee Comment (RC3) · Anonymous Referee #3 · 8 Apr 2018

**1   Overall remarks**

The paper reports on about three years of tropospheric ozone and formaldehyde measurements from a new FTIR instrument in Heifei, China. The data are compared to a number of correlative data, including tropospheric NO2 from the OMI satellite instrument, and results from chemical transport models.

The authors give a very long and detailed description of their instrument and retrieval

technique. They then analyse their observations using the correlative data mentioned above. Overall, their results, such as annual cycle, correlations, and trajectory analyses are plausible. However, the authors tend to discount differences and poor correlations, and to ignore the very coarse altitude resolution of their tropospheric ozone data, which average over a very wide altitude range, and have relatively little sensitivity to the planetary boundary layer, where a substantial part of the smog related ozone photo-chemistry takes place.

Largely I concur with the comments by the other two reviewers. The paper does not present major new insights. However, it is important to report on new instruments and on tropospheric chemistry findings in China. Therefore, and also considering that this is a special issue for the last Quadrennial Ozone Symposium, I recommend publication after a few major deficits have been addressed.

**2  Suggested Changes**

The description of the FTIR technique and FTIR profile retrieval in lines 148 to 248, as well as the averaging kernel smoothing used for comparison (lines 249 to 279) is pretty much standard. This could all be omitted, or moved to an appendix. A short paragraph and a few references in the main text are enough.

lines 339 to 342, lines 367 to 370: I think these simple attributions to "model input files" are not valid. The wide averaging kernels and low sensitivity of the FTIR tropospheric ozone columns to boundary layer ozone, as well as the limited horizontal resolution of the model data could play a very large role here. Please reword or omit these parts.

Appendix A: Basically this is textbook / Rogers (2000), right? So this could/ should be omitted.

Fig. 6: I am not sure how meaningful this comparison of ozone profiles is. Both

[Figure]

have very poor altitude resolution, and profile shape is determined to a very large degree by a priori assumptions. Comparison with a real tropospheric ozone profile from ozone-sondes or lidar would be much more meaningful. Maybe drop this Figure and its discussion? Similar considerations apply to Figs. 8 and 10.

In most respects, I concur with the detailed recommendations by the other two reviewers. However, after shortening, and addressing the major comments, I think this manuscript is publishable in ACP.

---

## Author Comment (AC1) · 22 May 2018

Response to short comments (SCs) from the scientific community:

Thanks very much for your comments. Our response to your comments are listed as follows. There is an extensive discussion among the authors regarding how to revise the content. So the response is delayed, and we are sorry for this.

The authors may wish to refer to recent work by Schroeder et al. (2017) doi: 10.1002/2017JD026781, who did a thorough analysis of the utility of column HCHO/NO2 ratios to indicate surface $O_3$. Schroeder et al. found that column HCHO/NO2 ratios are not effective for determining surface $O_3$ sensitivity, as the "transition" range is certainly larger than the range of 1-2 described in Duncan et al (2010), and larger than the 1.3-2.8 used in this work.

**Response:** Thanks very much for sharing your recent work, in which a wonderful and thorough analysis of the utility of column $HCHO/NO_2$ ratios to indicate surface $O_3$ is presented. We have referred to your work and made a brief introduction of your finding in the revised version. What we gained from your work is that the column $HCHO/NO_2$ ratios can be used to indicate surface $O_3$ sensitivity but special cares should be taken or, in other words, whether the column $HCHO/NO_2$ ratios can be used to indicate surface $O_3$ sensitivity depends on conditions. This is because many additional sources of uncertainty (regional variability, seasonal variability, variable free tropospheric contributions, retrieval uncertainty, air pollution levels and meteorological conditions) may cause transition threshold vary both geographically and temporally, and thus the results from one region are not likely to be applicable globally.

Tropospheric ozone is not an emission pollutant, but produced by photochemical oxidation of CO, $NO_x$, and VOCs under certain meteorological condition. This process is complicated and thus shows regional representativeness. We quite follows your idea, and actually, the transition thresholds either estimated by you or Sillman (1995) or Martin et al. (2004a) or Duncan et al. (2010) were not straightly applied in this work. Instead, after referring to previous work by you, Sillman (1995), Martin et al. (2004a) and Duncan et al. (2010), we use a compromise (conservative) way to estimate the transition threshold applicable in Hefei, China. That is, we iteratively

altered the HCHO/NO$_2$ ratio threshold and judged whether the correlations/sensitivities of tropospheric O$_3$ to HCHO and NO$_2$ changed abruptly. Specifically, in order to find out the VOC-limited threshold, we first fit tropospheric O$_3$ to tropospheric HCHO lie in HCHO/NO$_2$ < 2 (a start point), and write down the correlation and slope, then we alter the threshold by a step size of 0.1 and perform the fit again, i.e., only fit the data pairs that lies in HCHO/NO$_2$ < 1.9, and so on. Finally, we compared all correlations and slopes, and regarded the ratio that has a sudden change as the VOC-limited threshold. The method to find the NO$_X$-limited threshold is straightforward. It is reasonable based on a fact that ozone production is more sensitive to VOCs if it is VOCs-limited and is more sensitive to NOx if it is NOx limited, which is also a common sense among you work, Sillman (1995), Martin et al. (2004a), and Duncan et al. (2010). Thus, there should be a turnover point near the threshold. This technique has been used by Su et al. (2017, DOI:10.1038/s41598-017-17646-x) to investigate the O3-NOx-VOCs sensitivities during the 2016 G20 conference, and can indeed reflect the overall O$_3$ production conditions in one region.

Furthermore, we have chosen the same upper limits for all gases and they are not equivalent to the real tropopause heights, but are about 3 km lower than the mean value which, derived from the NCEP database, is 15.1 km with a standard deviation (1σ) of 1.1 km for Hefei. This manner not only ensured the accuracies of tropospheric O$_3$, CO, and HCHO retrievals, but also minimized the influence of transport from stratosphere, i.e., the so called STE process (stratosphere-troposphere exchange). On the other hand, we only selected the retrievals at the noon (around 13:35) for the O$_3$-NOx-VOCs sensitivities, which can minimized the uncertainty due to temporal variability.

**Related change:** We have referred to your work and made a brief introduction of your finding in the revised version.

---

## Author Response (AR1)

NOTE: This file includes two sections. Section 1 presents comments from referees, the corresponding point-by-point responses, and the related changes in the manuscript. Section 2 is the marked-up manuscript.

**Section 1:** (the black font are comments from referees, the red font are authors' responses as well as the related change clarifications.)**

**Comment response to all referees:**

Thanks very much for your comments, suggestions and recommendation with respect to improve our paper. The response to all your comments are listed below. There was an extensive discussion among the authors regarding how to revise the content, and this paper is subjected to a major revision including an update of all retrievals using new inputs (e.g.,  $S_a$  based on standard deviation of a dedicated WACCM run from 1980 to 2020), re-plot all figures, condense/reorganize the content and focus more on the scientific topics. Thus, the response is delayed, and we are sorry for this.

**(1) Detailed response to comments from referee #1:**

General comments :

The authors have used a new FTIR dataset to infer tropospheric ozone seasonal evolution and photochemical production regime at Hefei in China. Comparisons of the new dataset with OMI observations, and the GEOS-Chem and WRF-chem model data have shown good agreements. Back trajectories analyses have been used to attribute the contribution regions, and seasonal variabilities, to the high ozone levels observed at Hefei. The chemical sensitivity to ozone production has been studied at Hefei by using proxies such as CO and HCOH.

Although the authors use a new dataset, the novelty of some results is hard to admit.

For instance, it is presented the fact that tropospheric ozone column is higher in spring/summer as a key result, which is a known scientific idea (same for better agreements comparing smoothed profiles relative to unsmoothed profiles). A reorganization of the paper's structure is needed, with less focus on the know results and more thinking about what is the paper contribution to scientific progress. In addition, the goal of the comparisons of the new dataset with independent data (atmospheric models and satellite observations) is unclear, as well as the use of two

different model (global and regional). The objectives of the paper should be clarified and listed in a concise way. The number of figures should be reduced to fit the main scientific results. Discussions about results, such as model and observations comparisons, are missing and would improve the scientific impact of the paper. The thorough section describing the retrievals is well written and I would advise the authors to submit this paper to a more technical journal, such as Atmospheric Measurement Technique, if not addressing these comments.

**Response:** This paper has been subjected to a major revision based on the comments from three referees. All your comments are appreciated and have been addressed in the revised version. Main changes/improvements are listed as follows:

1) We have updated all retrievals with new  $S_a$  deduced from standard deviation of a dedicated WACCM run from 1980 to 2020, which should be more close to actual natural variation compared to the previous version. This improvement doesn't change the results of this paper.

**2**) We have reorganized the paper's structure, with less focus on known results and more describing about what is scientifically new. The objectives of the paper are clarified and listed in a concise way. The number of figures is reduced to focus more on the main scientific results. We have condensed quite a lot the descriptions of site/instrument, retrieval, theoretical basis but added many discussions/explanations regarding the observed results and photochemical regime. The figures and descriptions that are useful for understanding this paper but not scientific new are now shifted to the supplement (e.g., previous figures 2 - 5).

**3**) After an extensive discussion among the authors, we deleted all paragraphs and figures regarding comparisons with the correlative data, i.e., OMI, GEOS-Chem and WRF-Chem data, due to the following reasons:

a) The scientific topic of our manuscript is the investigation of the ozone seasonal evolution, source and photochemical production regime in polluted eastern China. The main interesting message we would like to present is the application of the FTS tools to determine if the tropospheric  $O_3$  is produced by NOx or VOC, and give a recommendation about what could be done to mitigate the high  $O_3$  levels. This can not

2

only improve the understanding of regional photochemical  $O_3$  production regime, but also contributes to the evaluation of  $O_3$  pollution controls. In the revised version, we leads straightly to this recommendation. For things which are not important for the main message, especially the deviation or something which probably misleads a potential reader, are removed. Accordingly, we removed the comparison with the models and the satellite.

b) This topic regarding comparisons with the correlative data, i.e., OMI, GEOS-Chem and WRF-Chem data, is interesting, but it cannot be clarified clearly within a few sentences or paragraphs and is basically a separate paper. Considering that this paper is already very long (referee's comments), we keep the intention of investigating the ozone seasonal evolution, source and photochemical production regime and removed all comparison with the correlative data.

**4**) We have responded to all referees' comments point-by-point and revised the manuscript accordingly.

**Related change:** The changes/improvements listed above have been done in the revised paper.

Specific comments:

Concerning the structure of the paper, it needs to be reorganized with a shorter abstract focusing a key results, more detailed introduction about the proxies used to assess the chemical sensitivity to ozone production, more sub-sections and tables summarizing the results of the comparisons, less figures, and appropriate English language.

**Response:** We have reorganized the paper's structure, shortened the abstract to focus on a key results, and included more detailed introduction about the proxies used to assess the chemical sensitivity to ozone production. In addition, more sub-sections and tables are used, and the number of figures are reduced to focus on the main scientific results. The revised paper has been corrected by a copy-editing service to improve the language.

**Related change:** The changes/improvements listed above have been done in the revised paper.

Consistency is also needed across the paper: define once (NDACC has never been defined in the abstract but appears in the keyword section, same for HCOH and VOC in the abstract, ...) and use them along the manuscript (ozone or  $O_3$ ?). Change old references with newest and avoid Wikipedia as a reference.

**Response:** All acronyms are now defined when they are first mentioned and also used consistently along the manuscript. Most old references are replaced with the newest ones and the Wikipedia reference is removed.

**Related change:** All these problem have been addressed in the revised paper.

In the introduction section, more explanations about why using proxies such as CO and HCOH would help the reader understanding how to assess the chemical sensitivity to ozone production.

**Response:** We have added more detailed introduction about the proxies used to assess the chemical sensitivity to ozone production in the introduction section, which would help the reader understanding how to assess the chemical sensitivity to ozone production.

**Related change:** See introduction in the revised paper.

In the retrieval strategy section, you mention a meteorological station onsite. Do you correct the NCEP profiles with these data? If yes, it should be clarified.

**Response:** As done at the other FTIR sites of the network, we did not correct the NCEP profiles with these data because this step normally makes the a priori profile (pressure, temperature) inconsistent. The pressure/temperature profiles have to obey some rules and this is fulfilled in the model data. The correction is also not that crucial, because the layers chosen depend only to a small extent on the temperature. When creating HDF files for the NDACC database, people usually have a field for surface temperature. But it is optional.

**Related change: None**

Define the use of the coincidence criteria when comparing to OMI (Why 3 hours and 0.7 ?) and other independent data.

**Response:** After an extensive discussion among the authors, we deleted all paragraphs and figures regarding comparisons with the correlative data, i.e., OMI, GEOS-Chem

and WRF-Chem data. Now this problem doesn't exist in the revised version. Please check above clarification (page 2) for the reason.

**Related change:** Please check the revised version for details.

In section 5.1, it is mentioned a trend analysis on a 4-years time series. The word

"trend" needs to be changed.

**Response:** This has been done in the revised version.

Related change: Please check section 4.1 in the revised version.

Are back trajectories used to investigate the regions of influence of high tropospheric ozone at Hefei? If yes, it needs to be clarified and better structured in a sub-section. The end of section 5.1 needs to be better structured to emphasis on the scientific conclusions.

**Response:** The back trajectories are used to determine the origin of the air masses. This has been clarified and the previous section has been re-structured into two sub-sections.

**Related change:** Please check section 4 for details.**

Concerning the comparisons with models, you may want to clarify their use; it is unclear if it is to assess the new dataset quality or investigate the model performances to reproduce observations. Explain the scientific interest of comparing the FTIR dataset with a global and a regional model. Discussions about results concerning comparisons between model/satellite and FTIR observations are missing and would raise the scientific level of the paper. Why is there a shift in the seasonal maximum in GEOS-Chem? Why do OMI and the FTS exhibit different seasonality? For WRF-Chem it is mentioned that the difference could be attributed to uncertainties in the input files, but what about the meteorological data, and/or the chemistry? This has to be further analyzed and explained.

**Response:** After an extensive discussion among the authors, we deleted all paragraphs and figures regarding comparisons with the correlative data, i.e., OMI, GEOS-Chem and WRF-Chem data. Now all these problems don't exist in the revised version. Please check above clarification (page 2) for the reason.

**Related change:** Please check the revised version for details.

Technical corrections :

- line 20 and 22 : define acronym CO, HCOH, and NO2

**Response:** We have defined these gases in the revised version.

**Related change:** Please check line 20 - 24 in the revised version.

- line 26 : by "the" FTS

- line27 : "occur" with no s

**Response:** This sentence has been removed when condensing the paper.

**Related change:** Please check the abstract in the revised version.

- line 32 and 33 : choose the precision, one or two decimal?

**Response:** Both are two decimal in the revised version.

**Related change:** Please check line 30 - 32 in the revised version.

- line 34 : by "atmospheric models" GEOS-Chem and WRF-Chem

**Response:** This sentence has been removed when condensing the paper.

**Related change:** Please check the abstract in the revised version.

- line 41-43 : rephrase the sentence

**Response:** We have rephrase it as "Compared with SON/DJF season, the observed tropospheric O3 levels in MAM/JJA are mainly influenced by transport of air masses from densely populated and industrialized areas while the broad and high O3 level and variability in MAM/JJA is determined by the photochemical O3 production." Please check abstract for details.

Related change: Please check line 32 - 35 in the revised version.

- line 45-50 : state that HCHO is a VOC and define VOC

**Response:** We state that HCHO is a VOC and define VOC in the revised version. Please check the second sentence in the abstract for details.

**Related change:** Please check line 20 - 23 in the revised version.

- key words : NDACC never defined in the abstract

**Response:** As far as we know, the key words part is not a mandatory part of ACP, and thus we have removed the key words part in the revised version. The definition for NDACC has been done in the main text (introduction).

**Related change:** Please check line 135 in the revised version.

- line 55 : add a reference

- line 56 : add a reference

**Response:** This has been removed when condensing the paper.

Related change: Please check line 44 in the revised version.

- line 53-71 : references are old

**Response:** Some old references have been replaced by the references published recently.

Related change: Please check line 44-70 in the revised version.

- line 75-77 : why so many references? Are they all relevant? You may use the most relevant one.

**Response:** This paragraph focuses on descriptions of the NDACC network. In the revised version, I removed the whole paragraph since it doesn't have much contributions to the main point of this paper. According, all references (if not referred in elsewhere) are also removed.

Related change: Please check introduction in the revised version.

- line 78-79 : define all chemical species.

- line 84 : state the accuracy or use a reference

**Response:** The whole paragraph has been removed, see above.

- line 87 : avoid Wikipedia as a scientific reference

**Response:** This reference has been replaced by two scientific papers.

**Related change:** Please check line 118 in the revised version.

- line 88 : first time ozone is written O3. Be consistent across the manuscript

**Response:** In the revised version, all "ozone" are replaced by " $O_3$ ". Now it is consistent across the paper.

- line 91 : "PM2.5"

- line 93 : "the" FTS

- line 96 : "Most NDACC sites"

**Response:** These have been done in the revised version.

**Related change:** Please check line 134-140 in the revised version.

- line 99 : Is the Hefei site a NDACC site? It is not clear here

**Response:** Hefei has ran both NDACC and TCCON conventions for more than 4 years, but is still a candidate site rather than an official one because of certain data publicity policy by Chinese government, and not because of the data quality. We are in progress to become an official TCCON site and we believe it will be also possible to be an official NDACC in near future.

**Related change:** Most site/instrument descriptions are removed and two reference are cited here.

- line 103 : add a reference for OMI

**Response:** A reference has been included in the revised version.

Related change: Please check line 138 in the revised version.

- line 105 : "the" site description

- line 107 : clarify the sentence "ozone related gases"
- line 114 : reference to Figure 1b
- line 114-115 : rephrase
- line 117 : clarify why it is an important region
- line 118 : add a reference
- line 119-120 : rephrase
- line 123-125 : rephrase
- line 125 : change "demonstrated" to "showed"

- line 126 : "typical observation day in August"

Response: All above related sentence has been removed when condensing the paper.

Most site/instrument descriptions can be found in our previous paper (Yuan et al.,2017;

Wei et al., 2017).

**Related change:** Please check section 2 in the revised version.

- line 121 : change "the same as" to "similar to"

- line 139 : define MIR
- line 143 : "for O3 measurements"

**Response:** These have been done in the revised version.

Related change: Please check section 2 in the revised version.

- line 144 : are you certain filters are used to avoid detector non-linearity? What about

signal to noise ratios?

**Response:** Filters are used for both, avoid detector non-linearity and improve the signal to noise ratios. However, this sentence has been removed when condensing the paper.

Related change: Please check section 2 in the revised version.

- line 148 : clarify "ozone related gases"

**Response:** This has been changed to "FTS retrievals of  $O_3$ , CO and HCHO" in the revised version.

Related change: Please check line 163 in the revised version.

- line 151 : how much is an adequate accuracy?

- line 152-155 : is this sentence misplaced ? If not explain why it "confirms"

- line 155 : delete "so"

- line 157 : delete "overpass"

**Response:** To avoid misunderstanding, this paragraph has been removed when condensing the paper. Accuracy estimation can be found in section 3.3. The whole section 3 is used to confirm tropospheric O3, CO and HCHO are robust in Hefei.

**Related change:** Please check section 3 in the revised version.

- line 161 : add a reference to SFIT4

**Response:** This has been done in the revised version.

**Related change:** Please check line 166 in the revised version.

- line 195 : use mathematical equations

**Response:** We have used mathematical equations in the revised version and shifted it to supplement.

**Related change:** Please check Supplement section A in the revised version.

- line 230 : "Figures 4 and 5"

**Response:** This has been done in the revised version. The two figures have been shifted to the supplement, now it is Figures S4 and S5.

**Related change:** Please check figures S4 and S5 in the revised version.

- line 252 : explain why two sets of models

- line 282 : how much is 0.7 in kilometers at Hefei?

- line 307 : add the GEOS-Chem general reference

- line 320 : what is the nearest grid in kilometer?

- line 345 : add a general reference for WRF-Chem

- line 347 : "Liu et al. (2016)"

- line 348 : "20 x 20 km"

- line 382-383 : add a reference

- line 395 : delete "global"

**Response:** After an extensive discussion among the authors, we deleted all paragraphs and figures regarding comparisons with the correlative data, i.e., OMI, GEOS-Chem and WRF-Chem data. Now all these problems don't exist in the revised version. Please check above clarification (page 2) for the reason.

**Related change:** Please check the revised version for details.

- line 430 : "air pollution"

- line 430-432 : rephrase the sentence

**Response:** This sentence has been replaced by many detailed explanations in the revised version.

**Related change:** Please check section 4.2 in the revised version for details.

- line 435 : state the percentage

**Response:** We have stated the percentage in the revised version.

**Related change:** Please check section 4.2 in the revised version for details.

- line 439 : "considering the fact"

**Response:** This sentence has been replaced by many detailed explanations in the revised version.

**Related change:** Please check section 4.2 in the revised version for details.

- line 441 : "air pollution"

**Response:** This has been done in the revised version.

**Related change:** Please check section 4.2 in the revised version.

- line 454 : "Obvious". Why it is obvious?

**Response:** After an extensive discussion among the authors, we deleted all paragraphs and figures regarding comparisons with the correlative data, i.e., OMI, GEOS-Chem

and WRF-Chem data. Now this problem doesn't exist in the revised version. Please check above clarification (page 2) for the reason.

**Related change:** Please check the revised version for details.

- line 465 : "not an emission pollutant" is not clear, rephrase

- line 466-467 : explain why the fact that it is complicated means that it shows regional representativeness?

**Response:** This sentence has been removed when condensing the paper.

**Related change:** Please check section 5.1 in the revised version.

- line 479 : "as a result", explain further the link between the two sentences

**Response:** Many explanations have been included in the revised version.

**Related change:** Please check section 5.1 in the revised version for details.

- line 485 : stay at present

**Response:** This has been done in the revised version.

**Related change:** Please check line 340 in the revised version for details.

- line 497 : how much are a good and an adequate correlation?

**Response:** In previous version, we regard it as good correlation if the correlation is higher than 0.6, and regard it as moderate correlation if the correlation lies in between 0.4 and 0.6. However, in the revised version, we only present the numbers and don't use the description such as "good" or "moderate" or "poor" to avoid controversy.

**Related change:** Please check the revised version for details.

- line 502 : "has taken"

**Response:** This sentence has been changed to "Sillman (1995a) and Tonnesen and Dennis (2000) found that in situ measurements of the HCHO/NO2 ratio could be used to diagnose local photochemical regimes." and shifted to introduction part.

**Related change:** Please check line 78 in the revised version for details.

- line 505-513 : this could go to the introduction section

**Response:** We have shifted these sentence to introduction section.

**Related change:** Please check line 95-105 in the revised version for details.

- line 525 : change "obtainment"

Response: Has been changed to "the measurement tool for HCHO in this study was

not the same as that of..."

Related change: Please check line 386 in the revised version for details.

- line 554 : change "validate" since OMI, GEOS-Chem, and WRF-Chem, to my knowledge, have already been validated

**Response:** After an extensive discussion among the authors, we deleted all paragraphs and figures regarding comparisons with the correlative data, i.e., OMI, GEOS-Chem and WRF-Chem data. Now this problem doesn't exist in the revised version. Please check above clarification (page 2) for the reason.

**Related change:** Please check conclusion in the revised version.

- Figure 1 a : change to see star colors

- Figure 1 b : instead of showing SZA for 1 day, you may want to show the mean SZA for all the data involved in this study

- Figure 1 legend : is it relevant to point out the wetlands? Are the red hexagons SZA or azimuth angle?

**Response:** In order to present the objectives of this paper in a concise way, the content has been shortened quite a lot. We removed this figure in the revised version. Detailed site/instrument descriptions can be found in our previous paper (Yuan et al., 2017; Wei et al., 2017).

**Related change:** Please check section 2 in the revised version for details.

- Figure 2 : cut altitude at 60 or 80 km

**Response:** This has been done in the revised version and already shifted to supplement.

Related change: Please check the caption of figure S2 for details.

- Figure 3 : arrange the figure so that the text is readable and is not crossed by the lines. Figure 3 partial column averaging kernel of HCOH: explain what are the influences on retrieved column of a partial Avk of 12.

**Response:** This has been done in the revised version and is shifted to supplement. For partial column averaging kernel of HCOH, we find a bug in our previous plotting script. In the revised version, we fixed this bug and now this problem doesn't exist. This bug has no influence on retrieval but on for PAVK plotting. Thus, every

deduction is the same.

Related change: Please check figure S3 for details.

- Figure 4 and 5 : cut at 60 or 80 km and combine them in one Figure

**Response:** Both have been cut at 60 km, but we did not combine them in one figure because there are so much error components, and the combination is a big mess. We have shifted them to the supplement, please check for details.

Related change: Please check figures S4 and S5 for details.

- Figure 6, 8, and 10 : insert the number of points included in the comparison and insert the standard deviation of the mean

- Figure 7 and 9 a : insert error bars

- Figure 7 b : shift text

- Figure 11 b : why showing both biased and unbiased data?

**Response:** After an extensive discussion among the authors, we deleted all paragraphs and figures regarding comparisons with the correlative data, i.e., OMI, GEOS-Chem and WRF-Chem data. Now this problem doesn't exist in the revised version. Please check above clarification (page 2) for the reason.

**Related change:** Please check the revised version for details.

- Figure 12 b : reduce y-axis scale

**Response:** This has been done in the revised version.

**Related change:** Please check figure 1b for details.

- Figure 15 : maybe plot all the measurements involved instead of daily means?

**Response:** Now all measurements were included in the revised version.

**Related change:** Please check figure 3 for details.

- Figure 16 : reduce the size of the dots. Do you consider error bars to fit the data?

**Response:** We have reduced the size of the dots and grouped them into different seasons in the revised version. The error bars were not included in the fit because the meteorological station data do not have uncertainties. We get the accuracy of each element from the user manual.

**Related change:** Please check figure 4 for details.**

- Figure 18 : (a) all 3 panels should fit in one page. (b) Do you account for

uncertainties when fitting the data? (c) R = 919 with 8 points, are you certain it is a robust statistic?

**Response:**

R(a): In the revise version, the 3 panels have been fitted in one page.

R(b): We account for both slope and correlation. Briefly, we iteratively altered the column HCHO/NO2 ratio threshold and judged whether the sensitivities of tropospheric O3 to HCHO or NO2 changed abruptly. For example, in order to estimate the VOC-limited threshold, we first fitted tropospheric O3 to HCHO that lies within column HCHO/NO2 ratios < 2 (an empirical start point) to obtain the corresponding correlation/slope, and then we decreased the threshold by 0.1 (an empirical step size) and repeated the fit, i.e., only fitted the data pairs with column HCHO/NO2 ratios < 1.9. This has been repeated. Finally, we sorted out the transition ratio which shows an abrupt change in correlation/slope, and regarded this as the VOC-limited threshold. Similarly, the NOx-limited threshold was determined by iteratively increasing the column HCHO/NO2 ratio threshold till the sensitivity of tropospheric O3 to NO2 changed abruptly.

R(c): The previous figure (R = 919 with 8 points) is only used to demonstrate that PO3 is more sensitive to VOC within VOC-limited region. Actually, the transition occurs close to about 0.6. At the transition ratio, there are much more points than 8. In the revised version, a detailed description for obtaining the transition threshold is presented, this kind of subfigures (only used for examples) are all removed.

**Related change:** Please check figure 6 for details

- Table 1 : enlarge the first column to adjust the word "regularization". O3668 with exponent and index

**Response:** This has been done in the revised version.

**Related change:** Please check table 1 for details.

**(2) Detailed response to comments from referee #2:**

Summary:

The authors report on solar absorption FTIR measurements of tropospheric columns

of O3, CO, and HCHO at a candidate NDACC IRWG observation station in Hefei, China. High spectral resolution measurements were conducted between 2014 and 2017 and fill a data gap within the NDACC observation network. The data shows higher tropospheric O3, also with higher variability, in spring and summer. The authors compare these O3 measurements to OMI satellite O3 (PROFOZ product), as well as GEOS-Chem (2 x 2.5 deg) and WRF-Chem (20 x 20 km) model O3 outputs. Comparisons are done in both profile and tropospheric partial column form.

Ozone FTS vs. GEOS-Chem model differences (481 coincidences) are attributed to uncertainties in GEOS-Chem input files ("ozone production loss rates and emission inventory"), it is concluded that GEOS-Chem is biased 13% lower (along profile), with r=0.5 for tropospheric column correlation plots.

Ozone FTS vs. WRF-Chem model differences (481 coincidences) are attributed to uncertainties in WRF-Chem input files ("ozone production and loss rates and MEIC inventory"), it is concluded that WRF-Chem is biased 12% lower (along profile), with r = 0.65 for tropospheric column correlation plots.

Comparisons to coincident OMI ozone profiles and partial (tropospheric) OMI columns were done on 53 coincident measurements after filtering for  $0.7^{\circ}$  spatial coincidence. Coincident FTS profiles were averaged in a 3 hour window around the OMI overpass at 13:30. OMI profiles were smoothed with FTIR averaging kernels. The OMI profiles are biased 2-13% lower than FTIR profiles, with r=0.73 for tropospheric column correlation plots, in which most OMI points sit below the 1:1 line, indicating also a low bias of OMI w.r.t. FTS.

Both sets of model ozone data are described as "smoother" than FTIR data and are "bias corrected" by adding a constant offset to the tropospheric  $O_3$  columns throughout the year to shift the model data towards FTIR partial column values. GEOS-Chem partial columns are increased by ~100% while WRF-Chem partial columns are increase by ~33% to increase agreement with FTS. Finally, OMI ozone partial column data were increased by ~20% and only then were monthly mean ozone partial column differences calculated.

24-hour back trajectories were calculated arriving at Hefei at 3000 m.a.s.l. from

15

2014-2017, presumably for those days with FTS observations (?), and they were grouped into spring/summer (presumably MAM/JJA?) and autumn/winter (presumably SON/DJF?). Summer transport is less vigorous and more varied than winter transport, as expected, bringing more air from highly polluted areas, e.g., east China, and broadly accounting for higher O3 and higher O3 variability in the data in spring/summer.

Finally, the O3 production regime is analyzed by describing correlations to meteorological variables recorded at Hefei, as well as looking at O3 vs. CO, O3 vs NO2 (for ratios of HCHO/NO2 > 2.8, assumed to correspond to NOx-ltd O3 production) and O3 vs HCHO (for ratios of HCHO/NO2 < 1.3, assumed to correspond to VOC-ltd O3 production). The ratios to indicate the O3 production regime were found iteratively until the correlation between O3 and NO2 or O3 and HCHO was > 0.6. 106 days of observations (O3, HCHO, CO from FTS; NO2 from OMI) were identified and of those 60% were NOx-ltd, 11% were VOC-ltd, and the remainder were mixed.

**Major comments:**

The paper is generally well written and presents a thorough error budget and sensitivity analysis of FTIR retrievals (O3, CO, HCHO) from a new candidate station in the NDACC network. The methods used here are well known and figures 2-5 should also move to the appendix, along with the Rodgers & Connor formulation, unless the authors highlight how their averaging kernels and error budget profiles differ from other similar published results. The paper presents a valuable new and growing observational dataset, however, this reviewer recommends major revisions in order to meet the ACP criteria of scientific significance and quality.

**Response:** This paper has been subjected to a major revision based on the comments from three referees. All your comments are appreciated and have been addressed in the revised version. Main changes/improvements are listed as follows:

1) We have updated all retrievals with new  $S_a$  deduced from standard deviation of a dedicated WACCM run from 1980 to 2020, which should be more close to actual natural variation compared to the previous version. This improvement doesn't change

the results of this paper.

**2**) We have reorganized the paper's structure, with less focus on known results and more describing about what is scientifically new. The objectives of the paper are clarified and listed in a concise way. The number of figures is reduced to focus more on the main scientific results. We have condensed quite a lot the descriptions of site/instrument, retrieval, theoretical basis but added many discussions/explanations regarding the observed results and photochemical regime. The figures and descriptions that are useful for understanding this paper but not scientific new are now shifted to the supplement (e.g., previous figures 2 - 5).

**3**) After an extensive discussion among the authors, we deleted all paragraphs and figures regarding comparisons with the correlative data, i.e., OMI, GEOS-Chem and WRF-Chem data, due to the following reasons:

a) The scientific topic of our manuscript is the investigation of the ozone seasonal evolution, source and photochemical production regime in polluted eastern China. The main interesting message we would like to present is the application of the FTS tools to determine if the tropospheric  $O_3$  is produced by NOx or VOC, and give a recommendation about what could be done to mitigate the high  $O_3$  levels. This can not only improve the understanding of regional photochemical  $O_3$  production regime, but also contributes to the evaluation of  $O_3$  pollution controls. In the revised version, we leads straightly to this recommendation. For things which are not important for the main message, especially the deviation or something which probably misleads a potential reader, are removed. Accordingly, we removed the comparison with the models and the satellite.

b) This topic regarding comparisons with the correlative data, i.e., OMI, GEOS-Chem and WRF-Chem data, is interesting, but it cannot be clarified clearly within a few sentences or paragraphs and is basically a separate paper. Considering that this paper is already very long (referee's comments), we keep the intention of investigating the ozone seasonal evolution, source and photochemical production regime and removed all comparison with the correlative data.

4) We have responded to all referees' comments point-by-point and revised the

17

manuscript accordingly.

**Related change:** The changes/improvements listed above have been done in the revised paper.

The FTS O3 measurements are higher than both models (global and regional) and the OMI measurements. The FTS measures a total column through a particular atmospheric slant column, and is expected to be less sensitive to local O3 events than an in situ sensor. We expect generally good agreement with downward-looking OMI, although coincidences are always a challenge. We also expect differences in the FTS vs. model comparisons because of different representativeness offered by a 20x20 km model vs. as  $2.5 \,^{\circ}x \, 2 \,^{\circ}$  model. This is not discussed in the paper.

Also, for the 20x20 km WRF-Chem model, the profile up to 10 km could extend over two horizontal grid boxes for most SZAs > 45 °, depending on the location of Hefei within a model grid box. Has this been considered?

Without discussing representativeness, the authors attribute FTS vs. model differences to model "input files", e.g., "ozone production loss rates and emission inventory" which is superficial. As a consequence, we learn little, if anything, about specific model processes and emission inventories that may be responsible.

Also, why is the data from this candidate station considered as "truth" in the comparison to OMI and the models? The total errors are estimated as 10% but they are dominated by smoothing error and based on very tight Sa values for  $O_3$  (10%), so (as the authors note), they are an underestimate.

If the authors plotted OMI vs. FTS trop  $O_3$  column data with both data sets' error bars they would still not overlap, but presumably OMI data has been validated – is it generally found to be low compared to other data?

The addition of a simple offset to model O3 values before looking at fractional monthly mean differences w.r.t. FTS is problematic because it is evident in figures 9 and 11 that such a simple manipulation does not bring the data points onto the 1:1 line. Instead, we have the highest O3 values below the FTS measurements and the lowest values above. This is even more dramatic in GEOS-Chem data, presumably because of lower model resolution, which homogenizes high  $O_3$  values over a large grid cell,

while raising the background O3 values.

Since the highest values occur in spring/summer and the lowest in autumn/winter, the bias is seasonally dependent and therefore not just due to spatial representativeness. Is it due to incorrect emissions or chemistry?

What are the main chemistry and emissions differences between the two models being compared to FTS? WRF-Chem is running with the MEIC inventory, presumably optimized for China, as well as biogenic emissions from MEGAN – why does it only do a little bit better than GEOS-Chem?

About smoothing the OMI profile by the FTIR averaging kernels, this method is meant to be applied to high vertical resolution correlative data, which OMI is not. It has about ~1 DOF in the troposphere itself. This may explain why there is still a lot of "shape" left in the fractional difference between FTIR and smoothed OMI profiles. What do OMI kernels look like and where is its peak of sensitivity – is it the same as for FTS?

**Response:** After an extensive discussion among the authors, we deleted all paragraphs and figures regarding comparisons with the correlative data, i.e., OMI, GEOS-Chem and WRF-Chem data. Now all these problems don't exist in the revised version. Please check above clarification (page 4) for the reason.

**Related change:** Please check the revised version for details.

The trajectory cluster analysis is difficult to follow without familiarity with China's geography. That can easily be fixed by adding the major city or region names referred to in the discussion to Figure 13. Without this information, it is hard to quickly judge if 1-day trajectories are long enough for transport to occur to Hefei. It is also not clear how the trajectories are clustered and the mean cluster trajectories (in color) are hard to see. Another way to represent this data would be to count trajectory elements crossing, e.g.,  $0.5 \circ x \ 0.5 \circ grid$  boxes. Also, why 3000 m? That seems much higher than the typical boundary layer height in winter, and probably also in summer. This choice will influence strongly both the speed and footprint of the pollution regions influencing Hefei. Have the authors tried 1500 m?

**Response:** In the revised version, all your comments regarding coincident trajectory

cluster analysis have been addressed. Now we used 1500 m a.s.l. While the relative contribution/direction of each trajectory changes a little bit, the main point is still the same.

**Related change:** Now the height is 1500m, and China's geography is included. Please check figure 2 in the revised version for details.

Finally, regarding O3 production regimes, ratios of HCHO/NO2 were varied until the correlation was > 0.6 in plots of O3 vs. HCHO and O3 vs. NO2. The outcome is that the correlation for the NOx-ltd plot of O3 vs. NO2 is 0.66 (moderate) while the correlation for the VOC-ltd plot of O3 vs. HCHO is 0.92, with far fewer points remaining in the fit. This seems rather arbitrary and needs justification. Also of the 106 days available for this analysis, which are from spring/summer and which are from autumn/winter? Are all VOC-ltd days in winter?

**Response:** a) The previous figure (R = 0.919 with 8 points) was only used to demonstrate that PO3 is more sensitive to VOC within VOC-limited region. Actually, the transition occurs close to about 0.6 (not 0.919). At the transition ratio, there are many more points than 8. In the revised version, a detailed description of obtaining the transition threshold is presented, this kind of subfigures (only used for demonstration) are all removed. Briefly, we iteratively altered the column HCHO/NO2 ratio threshold and judged whether the sensitivities of tropospheric  $O_3$  to HCHO or NO2 changed abruptly. For example, in order to estimate the VOC-limited threshold, we first fitted tropospheric  $O_3$  to HCHO that lies within column HCHO/NO2 ratios < 2 (an empirical start point) to obtain the corresponding correlation/slope, and then we decreased the threshold by 0.1 (an empirical step size) and repeated the fit, i.e., only fitted the data pairs with column HCHO/NO2 ratios < 1.9. This has been done iteratively. Finally, we sorted out the transition ratio which shows an abrupt change in correlation/slope, and regarded this as the VOC-limited threshold. Similarly, the NOx-limited threshold was determined by iteratively increasing the column HCHO/NO2 ratio threshold till the sensitivity of tropospheric  $O_3$  to  $NO_2$  changed abruptly.

The transition threshold estimation using this scheme exploits the fact that O3

20

production is more sensitive to VOCs if it is VOCs-limited and is more sensitive to  $NO_x$  if it is  $NO_x$  limited, and it exists a transition point near the threshold (Martin et al., 2004). Su et al. (2017) used this scheme to investigate the O3-NOx-VOCs sensitivities during the 2016 G20 conference in Hangzhou, China, and argued that this diagnosis of PO3 could reflect the overall O3 production conditions.

b) Table 4 and the last paragraph of section 5.3.2 present detailed description of classification for these 106 days measurements. Not all but ~ 75% VOC-ltd days are in winter.

**Related change:** This problem has been addressed in the revised version. Please check section 5.3 in the revised version for details.

When I look at the full O3 data in Figure 12, I wonder why there isn't a stronger signature of JJA O3 enhancements in Hefei? (Is it related to filtering out days affected by haze, App B?) Many high values seem to be in May, although the x-axis is hard to read and should really be changed to, Jan 1, June 1,etc., throughout the paper where dates are shown. Or possibly at boundaries between MAM, JJA, SON, DFJ, if these are the groupings for the seasons in the paper.

**Response:** a) Compared to other high resolution FTS sites, the  $O_3$  measurement in Hefei in JJA are very high, and we observed higher day-to-day variations in summer than other seasons. Vigouroux et al. (2015) studied  $O_3$  trends and variability with eight NDACC FTS stations that have a long-term time series of  $O_3$  measurements, namely, Ny-Ålesund (79 ° N), Thule (77 ° N), Kiruna (68 ° N), Harestua (60 ° N), Jungfraujoch (47 ° N), Izaña (28 ° N), Wollongong (34 ° S) and Lauder (45 ° S). All these stations were located in non-polluted or relatively clean areas. The results showed a maximum tropospheric column in spring at all stations except at Jungfraujoch which extended into summer. This is because the stratosphere troposphere exchange (STE) is most effective during late winter and spring (Vigouroux et al. 2015). We don't think there isn't a stronger signature of JJA  $O_3$  enhancements in Hefei is related to filtering criteria which are used to guarantee the data quality. It is most probably because the STE process is weaker in summer, though

photochemical  $O_3$  production is higher. Thus, tropospheric  $O_3$  (STE fraction plus photochemical production fraction) in JJA is not the highest.

b) "June" and "MAM, JJA, SON, DFJ" have been used in the revised version.

**Related change:** "June" and "MAM, JJA, SON, DFJ" have been used in the revised version. Please check figure 1(b) in the revised version for details.

Have the FTS partial columns been compared to in situ O3 monitors in Hefei to see if they also show enhancements in May/June 2015 and 2016? What about the low values in Jan 2015 and 2017 vs. the higher ozone in Jan 2016?

**Response:** We did not compared the FTS to in situ  $O_3$  data due to a lack of co-existing in situ  $O_3$  measurements. The  $O_3$  variations in Jan 2016 are higher compared to Jan 2015 and 2017, most probably because of higher air pollution.

**Related change: None**

Finally, the Pearson coefficient of 0.35 - 0.6 was taken to mean "moderately correlated" in this work. Typically moderate correlation is associated with values of 0.5 - 0.8, since the lower bound would mean that the model fit to the data explains only 25% of the variations in the data. At 0.35 that drops to only 12%.

**Response:** In previous version, we regard it as good correlation if the correlation is larger than 0.6, and regard it as moderate correlation if the correlation lies in between 0.4 and 0.6. However, in the revised version, we only present the numbers and don't use a description such as "good" or "moderate" or "poor".

**Related change:** Please check section 5 in the revised version for details.

Further detailed technical comments:

Fig. 1a: most names in this figure are illegible. Use a cleaner map to reduce clutter.

Fig. 1b: no red hexagons are visible, but I assume the red arc is the azimuth and the un-described yellow circles are the SZA.

**Response:** In order to focus on the main objectives, the content has been shortened quite a lot. We removed this figure in the revised version. Detailed site/instrument descriptions can be found in our previous paper (Yuan et al., 2017; Wei et al., 2017).

**Related change:** Please check section 2 in the revised version for details.

Fig. 2: what does "with measured ILS" mean in this caption? Is the ILS characterized

with linefit and then fixed in the retrievals, or are some ILS parameters still being retrieved? Why is there a loss of sensitivity to HCHO right at the surface? Is this a priori related?

**Response:** a) In sfit4, the ILS can be treated with three options: one is assuming an ideal ILS, two is retrieving the ILS together with the trace gas retrieval, and three is using the measured ILS. We regularly used a low-pressure HBr cell to monitor the instrumental line shape (ILS) of the instrument, and included the measured ILS in the retrieval.

**b**) It is not a priori related but a characteristic of the HCHO retrieval. The sensitivity at the ground is low because of the very weak absorption feature of HCHO. The spectral signature at the ground is very broad, thus in the presence of noise very indistinguishable from the features created by the interfering species. The previous figures 2 and 3 have been shifted to supplement. Now is figures S2 and S3.

**Related change:** Please check section 3.1 in the revised version for details.

Fig. 3: the HCHO trop column AK seems unhealthy for growing so far past 1 quickly above ~3 km, even if there is little HCHO there. What is the reason for this shape?

**Response:** We find a bug in our previous plotting script. In the revised version, we fixed this bug and now this problem doesn't exist. This bug only for PAVK plotting, and has no influence on retrieval. Thus, every deduction is the same as before. Thanks for point out this bug.

Related change: Please check figure S3 for details.

Fig. 4: What is the explanation for the peak in the CO error at around ~3 km?

**Response:** This is due to smoothing at around 3 km.

**Related change:** It has been shifted to supplement.

Fig. 5: Legend seems reversed for total random error and z shift for CO.

**Response:** We plot the three gases ( $O_3$ , CO, HCHO) using the same script, and after a careful check with our plotting script, we find there is no problem between the total random error and z shift for CO.

**Related change:** It has been shifted to supplement.

Fig. 9 and 11: it's hard to judge seasons with the date labels as presented. Also, why

do these figures not have the identical number of O3 data points if they are derived from the same data filtering applied to FTS data that is described in App B?

Fig. 14: is based on Fig 12, not 13 as the caption says. Again, what are the model process and inventory differences leading to this? Panel a) says smoothed model, but is OMI not also smoothed in this figure?

**Response:** After an extensive discussion among the authors, we deleted all paragraphs and figures regarding comparisons with the correlative data, i.e., OMI, GEOS-Chem and WRF-Chem data. Now all these problems don't exist in the revised version. Please check above clarification (page 3) for the reason.

**Related change:** Please check the revised version for details.

Fig. 15: The wind sensor appears to be installed in a poor location as the wind speed never exceeds 0.3 m/s or ~1 km/h! If that is the case, then the wind direction data is also spurious. That's too bad, because I wanted to see a plot of Hefei  $O_3$  vs. wind direction to see if  $O_3$  is higher when winds blow from the city.

**Response:** The weather station gives an output every 10 seconds, but the previous figure 15 only presents the daily average data that coincident with  $O_3$ . The wind direction and wind speed are **vectors**, thus, the averages are quite different compared to the short term data. The changing wind direction is the reason why the daily averaged wind speed seems never exceeds 0.3 m/s or ~1 km/h, and not because the wind sensor in a poor location. The figure 3 in the revised version, which presents minutely, hourly, daily, and monthly averaged data, illustrates the features better. For minutely- averaged data, the wind speed can exceed 6 m/s.

Wind direction is also important because it affects pollution transport, giving rise to high  $O_3$  in downwind locations (Wang et al., 2016). The city downtown locates in eastern of the observation site and the majority of the Chinese population lives in the eastern part of China, easterly winds (direction less than 180°) could generally transport more pollutants to the observe area than westerly winds (direction larger than 180°), resulting in a higher  $O_3$  level.

**Related change:** Minutely, hourly, daily, and monthly averaged data are included. Please check figure 3 for details. Fig 16: In spite of problems above, the highest  $O_3$  values occur for the lowest of the low wind speeds, pointing to the accumulation of local pollution. There is a "moderate" negative correlation between O3 and RH – why? We could learn more if these data were colored according to spring/summer and autumn/winter.

**Response:** The data are now color coded into spring/summer (MAM/JJA) and autumn/winter (SON/DJF) groups in figure 4 in the revised paper. We have fitted the minutely, hourly and daily average data with the coincident  $O_3$ , and all of them showed weak negative correlation between  $O_3$  and RH. Elevated  $O_3$  concentrations generally occurs on days with dry condition, low pressure and low winds in Hefei probably because these conditions favor the accumulation of  $O_3$  and its precursors.

**Related change:** Please check figure 4 in the revised paper.

Fig. 19: hard to judge seasons with x-axis labels. Panel b is based on data in panel a that does not seem to sample seasons evenly. This should be discussed.

**Response:** In the revised paper, we only present the time series of column HCHO/NO2 ratios (figure 7), and the detailed discussion for PO3 limitation is listed in table 4. The HCHO and O3 are not retrieved within the same spectra, which means a measurement day that has a robust HCHO retrieval does not always has a robust O3 retrieval, vice versa. The previous figure 19 (a) presents all days that have robust HCHO and NO2, and figure 19 (b) presents the days that have robust HCHO, O3 and NO2. The criteria in figure 19 (b) is more stricter than figure 19 (a), and thus seems don't sample seasons evenly.

**Related change:** The previous figure 19 (b) is removed in the revised version and the detailed discussion for  $PO_3$  limitation is listed in table 4. Please check figure 7 in the revised version for details.

Table 1: retrieved interfering gases  $\rightarrow$  as columns, I presume, except for H2O, as noted? Also, WM  $\rightarrow$ MW. I'm not sure what footnote b means, please clarify.

**Response:** The rows for  $H_2O$  has been deleted, WM is changed to MW. All footnotes has been removed because we think they are not necessary.

**Related change:** Please check table 1 for details.

Manuscript:

P1L74: sun spectra  $\rightarrow$  solar absorption spectra

**Response:** This paragraph focuses on descriptions of the NDACC network. In the revised version, I removed the whole paragraph since it doesn't have much contributions to the main point of this paper.

Related change: This paragraph has been removed.

P1L3: replace wiki reference with something from the many, many refereed papers on Chinese modernization and growing air pollution problems.

**Response:** This sentence has been removed in the revised version.

P4L89: what are China's AQ standards in ppb for long- and short-term exposure?

**Response:** Tropospheric  $O_3$  was already included in the new air quality standard as a routine monitoring component (http://www.mep.gov.cn, last access on 23 May 2018), where the limit for the maximum daily 8 h average (MDA8)  $O_3$  in urban and industrial areas is  $160\mu$ g/m3 (~ 75 ppbv at 273 K, 101.3 kPa).

**Related change:** Please check line 122 in the revised version for details.

P4L95: greatly contribute to ozone pollution controls  $\rightarrow$  contribute to the evaluation of O3 pollution controls

**Response:** This has been done in the revised version.

**Related change:** Please check line 140 in the revised version for details.

P4L117: ... after it is itself validated as an NDACC site and it moves from candidate to regular status.

P5L129: then increases  $\rightarrow$  then SZA increases

P5L129-133: what region influences the measurements depends on the azimuth of observation, yes, but also on the direction and wind speed pushing air masses above Hefei, especially for the lowest parts of the atmosphere. This could be significant when local pollution events are occurring as some events can be completely swept away from the FTS obs path.

**Response:** We agreed with your comment but we have removed these descriptions when condensing the paper. Detailed site/instrument descriptions can be found in our previous paper (Yuan et al., 2017; Wei et al., 2017).

**Related change:** Please check section 2 for details.

P6L173: cited references missing from references section

**Response:** This problem has been addressed in the revised version.

**Related change:** Please check the reference section.

P6L178: please explain deweighting more clearly. What are instrument SNR levels without deweighting?

**Response:** This sentence has been removed when condensing this paper. The standard deweighting stuff can be found in NDACC network or on request from the co-author Mathias Palm who is one of the SFIT4 developer. The instrument SNR level without deweighting is around 200 to 600.

Related change: Please check section 3 for details.

P7L187: how are the Sa diagonal element magnitudes chosen? WACCM?

**Response:** In the revised version, Sa diagonal element is based on standard deviation of WACCM simulations from 1980 to 2020.

Related change: Please check section 3 for details.

P7L191: is the ILS retrieved in all retrievals or is it done with LINEFIT and then held constant?

**Response:** We normally perform cell measurement once per month. For all measurements within this month, it is done with LINEFIT and then held constant. We included this clarification in the revised version.

Related change: Please check section 3 for details.

P11L315-317: tagged O3 runs are mentioned, which would be nice and would allow the attribution of pollution to various source regions, but these 3 lines are very unclear (i.e., also about restart files)

P14L393-4: basically reproduced ... but with slight shifts in timing July is wrong in both models; why are they low in August? When is the local Hefei smog season?

**Response:** In the revised version, these problems don't exist because we removed all comparisons with correlative data. By the way, in my impression, most smog occurs in winter season.

**Related change:** Please check section 4 for details.

P14L403: Logan (1985) "observed"  $\rightarrow$  I presume this is a model study?

**Response:** This reference which based on both model simulation and observation has been replaced by some newly references in the revised version.

Related change: Please check section 4.2 for details.

P16L448: basically consistent throughout all seasons  $\rightarrow$  it really doesn't look like that to me; would be easier to think about if time series started with MAM as opposed to JFM.

P16L457-63: this really is a shallow explanation of what may be causing the differences, from which we learn nothing concrete. Also, how does larger air pollution increase uncertainty in either emission inventories or the photochemical regime? Isn't the latter, especially, something that is diagnosed from the emission rates and relative abundances of NOx and VOCs?

**Response:** After an extensive discussion among the authors, we deleted all paragraphs and figures regarding comparisons with the correlative data, i.e., OMI, GEOS-Chem and WRF-Chem data. Now all these problems don't exist in the revised version. Please check above clarification (page 3) for the reason.

Related change: Please check the revised version for details.

P17L495/6: which emissions are being discussed here: biogenic? anthropogenic? What are the expected magnitudes and timing of each?

**Response:** In the revised version, this problem doesn't exist because this sentence has been changed to "Pronounced tropospheric CO and  $NO_2$  variations were observed but the seasonal cycles are not evident probably because of air pollution which is not constant over season or season dependent".

**Related change:** Please check section 5.2 for details.

P18L528: straightly applied  $\rightarrow$  straight forwardly applied

**Response:** This has been done in the revised version.

**Related change:** Please check line 389 for details.

P19L554: "validate OMI"  $\rightarrow$  that's a strong statement given the unproven nature of these particular FTIR measurements, and given there's no reference to other OMI validation efforts and what they have typically revealed.

P19L560: WRF-Chem agreement is "better"  $\rightarrow$  it has a lower "bias" but greater summer differences. It's not clear if that is better given it is a high res model using optimized emissions for China

**Response:** After an extensive discussion among the authors, we deleted all paragraphs and figures regarding comparisons with the correlative data, i.e., OMI, GEOS-Chem and WRF-Chem data. Now all these problems don't exist in the revised version. Please check above clarification (page 3) for the reason.

Related change: Please check the revised version for details.

P22 L651: would not screening out hazy days eliminate a lot of JJA  $O_3$  pollution days? Haze isn't a problem for FTIR as much as non-constant intensity (e.g., clouds floating by during a ~20 minute observation time).

**Response:** This criterion that is used to eliminate bad spectra requires that the solar intensity variation (SIV) is less than 10%. Empirically, most of the variations are caused by floating clouds and some of them may be caused by other objects such as smog or unknown opaque object. The 10% empirical threshold keeps a reliable retrieval. We don't think it eliminated a lot of JJA O3 pollution days. Haze is not a key factor that cause the variation and we have removed the word "haze" in the revised version.

**Related change:** Please check supplement for details.

**(3) Detailed response to comments from referee #3:**

**1 Overall remarks**

The paper reports on about three years of tropospheric ozone and formaldehyde measurements from a new FTIR instrument in Heifei, China. The data are compared to a number of correlative data, including tropospheric NO2 from the OMI satellite instrument, and results from chemical transport models.

The authors give a very long and detailed description of their instrument and retrieval technique. They then analyse their observations using the correlative data mentioned above. Overall, their results, such as annual cycle, correlations, and trajectory analyses are plausible. However, the authors tend to discount differences and poor correlations, and to ignore the very coarse altitude resolution of their tropospheric

ozone data, which average over a very wide altitude range, and have relatively little sensitivity to the planetary boundary layer, where a substantial part of the smog related ozone photo-chemistry takes place. Largely I concur with the comments by the other two reviewers. The paper does not present major new insights. However, it is important to report on new instruments and on tropospheric chemistry findings in China. Therefore, and also considering that this is a special issue for the last Quadrennial Ozone Symposium, I recommend publication after a few major deficits have been addressed.

**Response:** This paper has been subjected to a major revision based on the comments from three referees. All your comments are appreciated and have been addressed in the revised version. Main changes/improvements are listed as follows:

1) We have updated all retrievals with new  $S_a$  deduced from standard deviation of a dedicated WACCM run from 1980 to 2020, which should be more close to actual natural variation compared to the previous version. This improvement doesn't change the results of this paper.

2) We have reorganized the paper's structure, with less focus on known results and more describing about what is scientifically new. The objectives of the paper are clarified and listed in a concise way. The number of figures is reduced to focus more on the main scientific results. We have condensed quite a lot the descriptions of site/instrument, retrieval, theoretical basis but added many discussions/explanations regarding the observed results and photochemical regime. The figures and descriptions that are useful for understanding this paper but not scientific new are now shifted to the supplement (e.g., previous figures 2 - 5).

**3**) After an extensive discussion among the authors, we deleted all paragraphs and figures regarding comparisons with the correlative data, i.e., OMI, GEOS-Chem and WRF-Chem data, due to the following reasons:

a) The scientific topic of our manuscript is the investigation of the ozone seasonal evolution, source and photochemical production regime in polluted eastern China. The main interesting message we would like to present is the application of the FTS tools to determine if the tropospheric  $O_3$  is produced by NOx or VOC, and give a

recommendation about what could be done to mitigate the high  $O_3$  levels. This can not only improve the understanding of regional photochemical  $O_3$  production regime, but also contributes to the evaluation of  $O_3$  pollution controls. In the revised version, we leads straightly to this recommendation. For things which are not important for the main message, especially the deviation or something which probably misleads a potential reader, are removed. Accordingly, we removed the comparison with the models and the satellite.

b) This topic regarding comparisons with the correlative data, i.e., OMI, GEOS-Chem and WRF-Chem data, is interesting, but it cannot be clarified clearly within a few sentences or paragraphs and is basically a separate paper. Considering that this paper is already very long (referee's comments), we keep the intention of investigating the ozone seasonal evolution, source and photochemical production regime and removed all comparison with the correlative data.

**4**) We have responded to all referees' comments point-by-point and revised the manuscript accordingly.

**Related change:** The changes/improvements listed above have been done in the revised paper.

**Response to** "the authors tend to discount differences and poor correlations, and to ignore the very coarse altitude resolution of their tropospheric ozone data, which average over a very wide altitude range, and have relatively little sensitivity to the planetary boundary layer, where a substantial part of the smog related ozone photo-chemistry takes place."

**Briefly**: Many scientists have proved that column technique (OMI, GOME, or airborne results) can be used to investigate PO3 sensitivity (Martin et al. 2004a; Duncan et al. 2010; Choi et al., 2012; Witte et al., 2011; Jin and Holloway, 2015; Mahajan et al., 2015; Schroeder et al., 2017; Jin et al., 2017). The NO2 used in this study is the same as most previous studies, the sensitivity/resolution of FTS O3 is close to that of OMI (Liu et al., 2010), the FTS HCHO is verified to be robust in troposphere in view of future satellite validation (Vigouroux et al., 2018). Thus, column technique used in this study is reasonable. We do acknowledge the paper by

Schroeder et.al. (2017) which was published during the preparation of the manuscript. Schroeder et.al. (2017) question the usability of the column technique to infer PO3 sensitivity. However, this manuscript does take into account much of the criticism mentioned by Schroeder et.el (2017): we calculated the transition thresholds with the measurements in Hefei rather than straightly applied the thresholds estimated by either previous studies. The FTIR measurements have a much smaller footprint than the satellite measurements. Also we concentrate on measurements recorded during midday, when the mixing layer has largely been dissolved. And furthermore, the measurements are more sensitive to the lower parts of the troposphere, which can be inferred from the normalized AVK's. This reason is simply, that the AVK's show the sensitivity to the column, but the column per altitude decreases with altitude.

In detail: Over polluted areas, both HCHO and tropospheric NO2 have vertical distributions that are heavily weighted toward the lower troposphere, indicating that tropospheric column measurements of these gases are fairly representative of near surface conditions. Many studies have taken advantage of these favorable vertical distributions to investigate surface emissions of NOx and VOCs from space (Boersma et al., 2009; Martin et al., 2004a; Millet et al., 2008; Streets et al., 2013). Martin et al. (2004a) and Duncan et al. (2010) used satellite measurements of column HCHO/NO2 ratio to explore near-surface O3 sensitivities from space and disclosed that this diagnosis of O3 production rate (PO3) is consistent with previous finding of surface photochemistry. Witte et al. (2011) used a similar technique to estimate changes in PO3 to the strict emission control measures (ECMs) during Beijing Summer Olympic Games period in 2008. Recent papers have applied the findings of Duncan et al. (2010) to observe O3 sensitivity in other parts of the world (Choi et al., 2012; Witte et al., 2011; Jin and Holloway, 2015; Mahajan et al., 2015; Schroeder et al., 2017; Jin et al., 2017).

**Related change:** Several references where the column technique (OMI, GOME, or airborne results) is used to investigate  $PO_3$  sensitivity have been included in the revised version. Please check the introduction part for details.

2 Suggested Changes

The description of the FTIR technique and FTIR profile retrieval in lines 148 to 248, as well as the averaging kernel smoothing used for comparison (lines 249 to 279) is pretty much standard. This could all be omitted, or moved to an appendix. A short paragraph and a few references in the main text are enough.

**Response:** In the revised version, the previous lines 148 to 248 have been condensed dramatically. The previous figures 2 - 5 have been shifted to the supplement. Please check section 3 for details. We still keep some of them because this paper is the first time to present  $O_3$ , HCHO, and CO time series at Hefei site. We think a brief introduction regarding site/retrieval setting/error analysis is useful. The previous lines 249 to 279 are all removed when condensing this paper.

Related change: Please check section 3 for details.

lines 339 to 342, lines 367 to 370: I think these simple attributions to "model input files" are not valid. The wide averaging kernels and low sensitivity of the FTIR tropospheric ozone columns to boundary layer ozone, as well as the limited horizontal resolution of the model data could play a very large role here. Please reword or omit these parts.

**Response:** After an extensive discussion among the authors, we deleted all paragraphs and figures regarding comparisons with the correlative data, i.e., OMI, GEOS-Chem and WRF-Chem data. Now this problem doesn't exist in the revised version. Please check above clarification (page 2) for the reason.

**Related change:** Please check the revised version for details.

Appendix A: Basically this is textbook / Rogers (2000), right? So this could/ should be omitted.

**Response:** Appendix A is a textbook stuff but useful for understanding this paper. It has been shifted to the supplement in the revised paper.

**Related change:** Please check Supplement section A in the revised version.

Fig. 6: I am not sure how meaningful this comparison of ozone profiles is. Both have very poor altitude resolution, and profile shape is determined to a very large degree by a priori assumptions. Comparison with a real tropospheric ozone profile from ozone-sondes or lidar would be much more meaningful. Maybe drop this Figure and its discussion? Similar considerations apply to Figs. 8 and 10.

**Response:** After an extensive discussion among the authors, we deleted all paragraphs and figures regarding comparisons with the correlative data, i.e., OMI, GEOS-Chem and WRF-Chem data. Now this problem doesn't exist in the revised version. Please check above clarification (page 2) for the reason.

**Related change:** Please check the revised version for details.

In most respects, I concur with the detailed recommendations by the other two reviewers. However, after shortening, and addressing the major comments, I think this manuscript is publishable in ACP.

**Response:** We have reorganized the paper's structure, with less focus on known results and more describing about what is the paper contribution to scientific progress. We have responded to all referees' comments point-by-point and revised the content accordingly. Thanks very much for your recommendation.

**Section 2: marked up file, as follows**

In brief, our paper has been subjected to a major revision based on the comments from three referees. In the marked up file, we only point out the main changes rather all revisions to avoid a big mess. Main changes/improvements are listed as follows: 1) We have updated all retrievals with new Sa deduced from standard deviation of a dedicated WACCM run from 1980 to 2020.

**2**) We have reorganized the paper's structure, with less focus on known results and more describing about what is scientifically new. The objectives of the paper are clarified and listed in a concise way. The number of figures is reduced to focus more on the main scientific results. We have condensed quite a lot the descriptions of site/instrument, retrieval, theoretical basis but added many discussions/explanations regarding the observed results and photochemical regime. The figures and descriptions that are useful for understanding this paper but not scientific new are now shifted to the supplement (e.g., previous figures 2 - 5).

3) We deleted all paragraphs and figures regarding comparisons with the correlative

data, i.e., OMI, GEOS-Chem and WRF-Chem data.

The marked up file is as follow, please check the red underlined sentences for details:

**Ozone seasonal evolution and photochemical production regime in polluted troposphere in eastern China derived from high resolution FTS observations**

Youwen Sun 1, 2)#, Cheng Liu 2, 3, 1)#1, Mathias Palm 4), Corinne Vigouroux 5), Justus Notholt 4), Qihou Hu 1), Nicholas Jones 6), Wei Wang 1), Wenjing Su 3), Wenqiang Zhang 3), Changong Shan 1), Yuan Tian 1), Xingwei, Xu 1), Martine De Mazi ère 5), Minqiang Zhou 5) and Jianguo Liu 1)

(1 Key Laboratory of Environmental Optics and Technology, Anhui Institute of Optics and Fine Mechanics, Chinese Academy of Sciences, Hefei 230031, China)

(2 Center for Excellence in Urban Atmospheric Environment, Institute of Urban Environment, Chinese Academy of Sciences, Xiamen 361021, China)

(3 University of Science and Technology of China, Hefei, 230026, China)

(4 University of Bremen, Institute of Environmental Physics, P. O. Box 330440, 28334 Bremen, Germany)

(5 Royal Belgian Institute for Space Aeronomy (BIRA-IASB), Brussels, Belgium)
(6 School of Chemistry, University of Wollongong, Northfields Ave, Wollongong, NSW, 2522, Australia )

**These two authors contributed equally to this work**

**Abstract:**

The seasonal evolution of  $O_3$  and its photochemical production regime in a polluted region of eastern China between 2014 and 2017 has been investigated using different observations and modelling. We used tropospheric ozone (O3), carbon monoxide (CO) and formaldehyde (HCHO, a marker of VOCs (volatile organic compounds)) partial columns derived from high resolution Fourier transform spectrometry (FTS), tropospheric nitrogen dioxide (NO2, a marker of NOx (nitrogen oxides)) partial column deduced from Ozone Monitoring Instrument (OMI), surface

Correspondence to: Cheng Liu (chliu81@ustc.edu.cn) or Youwen Sun (ywsun@aiofm.ac.cn)

meteorological data, and a back trajectory cluster analysis technique. A broad  $O_3$ maximum during both spring and summer (MAM/JJA) is observed; the day-to-day variations in MAM/JJA are generally larger than those in autumn and winter (SON/DJF). Tropospheric  $O_3$  columns in June are, on average,  $0.50 \times 10^{18}$ molecules\*cm-2 (47.6%) higher than those in December which has a mean value of  $1.05 \times 10^{18}$  molecules\*cm-2. Compared with SON/DJF season, the observed tropospheric O3 levels in MAM/JJA are mainly influenced by transport of air masses from densely populated and industrialized areas while the broad and high O3 level and variability in MAM/JJA is determined by the photochemical O3 production. The tropospheric column HCHO/NO2 ratio is used as a proxy to investigate the photochemical  $O_3$  production rate (PO3). The results show that the PO3 is mainly nitrogen oxides (NOx) limited in MAM/JJA, while it is mainly VOC or mix VOC-NOx limited in SON/DJF. Statistics show that NOx limited, mix VOC-NOx limited, and VOC limited PO3 accounts for 60.1%, 28.7%, and 11%, respectively. Considering most of PO3 are NOx limited or mix VOC-NOx limited, reductions in NOx would reduce most of the O3 pollution in eastern China.

**1** Introduction**

Human health, terrestrial ecosystems, and materials degradation are impacted by poor air quality resulting from high photochemical ozone (O3) levels (Wennberg and Dabdub, 2008; Edwards et al., 2013; Schroeder et al., 2017). In polluted areas, tropospheric O3 generates from a series of complex reactions in the presence of sunlight involving carbon monoxide (CO), nitrogen oxides (NOx  $\equiv$  NO (nitric oxide) + NO2 (nitrogen dioxide)), and volatile organic compounds (VOCs) (Oltmans et al., 2006; Schroeder et al., 2017). Briefly, VOCs first react with the hydroxyl radical (OH) to form a peroxy radical (HO2 + RO2) which increases the rate of catalytic cycling of NO to NO2. O3 is then produced by subsequent reactions between HO2 or RO2 and NO that lead to radical propagation (via subsequent reformation of OH). Radical termination proceeds via reaction of OH with NOx to form nitric acid (HNO3) (reaction (1), referred to as LNOx) or by radical-radical reactions resulting in stable

peroxide formation (reactions (2) – (4), referred to as LROx, where  $ROx \equiv RO_2 + HO_2$ ) (Schroeder et al., 2017):

| $OH + NO_2 \rightarrow HNO_3$                                                        | (1) |  |
|--------------------------------------------------------------------------------------|-----|--|
| $\underline{2HO_2 \rightarrow H_2O_2 + O_2}$                                         | (2) |  |
| $\underline{\mathrm{HO}_2 + \mathrm{RO}_2} \rightarrow \mathrm{ROOH} + \mathrm{O}_2$ | (3) |  |
| $\underline{2RO_2} \rightarrow ROOR + O_2$                                           | (4) |  |

Typically, the relationship between these two competing radical termination processes (referred to as the ratio  $LRO_x/LNO_x$ ) can be used to evaluate the photochemical regime. In high-radical, low-NOx environments, reactions (2) – (4) remove radicals at a faster rate than reaction (1) (i.e.,  $LROx \gg LNOx$ ), and the photochemical regime is regarded as "NOx limited". In low-radical, high-NOx environments the opposite is true (i.e.,  $LROx \ll LNOx$ ) and the regime is regarded as "VOC limited". When the rates of the two loss processes are comparable ( $LNOx \approx LROx$ ), the regime is said to be at the photochemical transition/ambiguous point, i.e., mix VOC-NOx limited (Kleinman et al., 2005; Sillman et al., 1995a; Schroeder et al., 2017).

Understanding the photochemical regime at local scales is a crucial piece of information for enacting effective policies to mitigate O3 pollution (Jin et al., 2017; Schroeder et al., 2017). In order to determine the regime, the total reactivity with OH of the myriad of VOCs in the polluted area has to be estimated (Sillman, 1995a; Xing et al., 2017). In the absence of such information, the formaldehyde (HCHO) concentration can be used as a proxy for VOC reactivity because it is a short-lived oxidation product of many VOCs and is positively correlated with peroxy radicals (Schroeder et al., 2017). Sillman (1995a) and Tonnesen and Dennis (2000) found that in situ measurements of the ratio of HCHO (a marker of VOCs) to NO2 (a marker of NOx) could be used to diagnose local photochemical regimes. Over polluted areas, both HCHO and tropospheric NO2 have vertical distributions that are heavily weighted toward the lower troposphere, indicating that tropospheric column measurements of these gases are fairly representative of near surface conditions. Many studies have taken advantage of these favorable vertical distributions to investigate surface emissions of NOx and VOCs from space (Boersma et al., 2009; Martin et al., 2004a; Millet et al., 2008; Streets et al., 2013). Martin et al. (2004a) and Duncan et al. (2010) used satellite measurements of column HCHO/NO2 ratio to explore near-surface  $O_3$  sensitivities from space and disclosed that this diagnosis of  $O_3$  production rate (PO3) is consistent with previous finding of surface photochemistry. Witte et al. (2011) used the similar technique to estimate changes in PO3 to the strict emission control measures (ECMs) during Beijing Summer Olympic Games period in 2008. Recent papers have applied the findings of Duncan et al. (2010) to observe  $O_3$ sensitivity in other parts of the world (Choi et al., 2012; Witte et al., 2011; Jin and Holloway, 2015; Mahajan et al., 2015; Jin et al., 2017).

With in situ measurements, Tonnesen and Dennis (2000) observed a radical-limited environment with HCHO/NO2 ratios < 0.8, a NOx-limited environment with HCHO/NO2 ratios >1.8, and a transition environment with HCHO/NO2 ratios between 0.8 and 1.8. With 3-d chemical model simulations, Sillman (1995a) and Martin et al. (2004b) estimated that the transition between the VOC- and NOx-limited regimes occurs when the HCHO/NO2 ratio is  $\sim$  1.0. With a combination of regional chemical model simulations and the Ozone Monitoring Instrument (OMI) measurements, Duncan et al. (2010) concluded that O3 production decreases with reductions in VOCs at column HCHO/NO2 ratio < 1.0 and NOx at column HCHO/NO2 ratio > 2.0; both NOx and VOCs reductions decrease O3 production when column HCHO/NO2 
[revised manuscript text omitted]

---

## Referee Report (RR2)

Paper: Ozone seasonal evolution and photochemical production regime in polluted troposphere in eastern China derived from high resolution FTS observations

Authors: Sun, Youwen, et al.

Manuscript: acp-2017-1176

The paper documents retrievals of ozone, formaldehyde, and carbon monoxide using FTS observations in the central portion of eastern China over the 2014-2017 period. The paper also provides some reasonable analysis of the sources of the ozone values at this station.

I read the revised version, and went back and looked over the original version. First, the authors have put in a nice effort into revising their manuscript and mainly responding to the critiques of the original manuscript. Second, the discussion on the retrievals, kernels, uncertainties is well done. I would rate my confidence in the data and retrievals to be quite high. Third, tropospheric chemistry in not my forte, but I'm skeptical of inferring information from column observations. Since ozone production is a non-linear function of VOCs and NOx, and since much of this is found in the boundary layer or strong plumes in the free troposphere, it is difficult to inter much information from column observations. Hence, I find some of the correlations shown in the manuscript to be not well founded or poor. Hence, the overall analysis is marginal in my "dynamicist" view. Nevertheless, this is interesting data from a highly polluted region that should be in the literature. Hence, I feel this should be published.

---

## Author Response (AR3)

**NOTE: This file includes two sections. Section 1 presents comments from the co-editor, referees, the corresponding point-by-point responses, and the related changes in the manuscript. Section 2 is the marked-up manuscript.**

**Section 1: (the black font are comments from the co-editor, referees, the red font are authors' responses as well as the related change clarifications.)**

**(1) Response to co-editor:**

Dear authors, the reviewers have made some recommendations for further minor revisions and corrections, please see the reports for details. Overall, the reviewers have commended you for the revisions already implemented and once the minor revisions are taken care of, we can proceed to publish the manuscript.

**Response:** This paper has been subjected to a minor revision/correction based on the comments from three referees. Detailed point-by-point response and the related changes are listed as follows:

**(2) Detailed response to comments from referee #1:**

Following the referee's remarks, the revised version has been strongly reorganized as compared to the initial version. As the authors state, (1) they have updated all retrievals with new Sa deduced from standard deviation of a dedicated WACCM run from 1980 to 2020, (2) they have organized the paper's structure, with focus on new results, and retaining only the most pertinent figures, (3) they omitted comparisons with the correlative data, i.e., OMI, GEOS-Chem and WRF-Chem data, (4) the reoriented the papers focus on photochemical ozone regime. In my view, this responded to the major referee remarks, and makes the paper rather different from the initial one. I think, the paper shows now interesting results on data at a specific site and an interesting discussion of the ozone formation regime that can be deduced from the data. However, there is a major difficulty, because from reading the paper I think that the O3 product is not specific for PBL $O_3$, which would be needed for the current analysis. So I would think the paper is in principle worthwhile for publication in ACP, but only if this major issue is resolved or correctly addressed. Also figures should be better explained and a general rereading by a native speaker would be needed.

**Response:** Thanks very much for your recommendation with respect to publish this paper in ACP. Detailed explanations for your mentioned major issue are present, and we have tried our best to improve the language problem. I hope the ACP's copy and language editing service can fix the rest if we did not find out all of them.

In Section 3, it is not clear to me, what exactly are the altitude ranges of the partial columns that are the basis for the paper, and whether they are for the case of ozone pertinent, that is focusing on lower tropospheric ozone.

For instance, it is stated in section 3: "In this study, we have chosen the same upper limit for the tropospheric columns for all gases, which is about 3 km lower than the mean value of the tropopause (~15.1 km)." If I understand right, this means that 0 – 12 km partial columns are considered. But looking at figures S2 and S3, such columns would be sensitive to $O_3$ values up to 15 km. Following figure S3, wouldn't it be wiser to analyse a 0 – 8 km partial column being most sensitive between 0 and 6-8 km? This would allow being more sensitive to lower tropospheric ozone.

**Response:** Yes, for all gases, the tropospheric columns considered here are based on 0-12 km integration. In this way we ensured the accuracies for the tropospheric $O_3$, CO, and HCHO retrievals, and minimized the influence of transport from stratosphere, i.e., the so called STE process (stratosphere-troposphere exchange). For $O_3$ and CO, as Figure S3 shown, it is wiser to analyse the 0 – 9 km partial column, which can reduce more STE influence and don't reduce much accuracies. However, for $NO_2$ and HCHO, the degrees of freedoms (DOFS) within 0-9 km is less than 1, and large uncertainty may arise. When taken everything into account, we selected 0 – 12 km partial column for all gases. On the other hand, as the following Figure 1 shows, there isn't much sensitivity difference between the 0-9 and 0-12km, and 0 – 12 km still holds most sensitivity on the lower tropospheric ozone. Actually, most previous studies even chosen the tropopoause as the upper limit (Duncan et al., 2010; Choi et al., 2012; Witte et al., 2011; Jin and Holloway, 2015; Mahajan et al., 2015; Jin et al., 2017).

**Related change:** None

[Figure]

Figure1. Partial column averaging kernels (PAVK) (ppmv / ppmv) for $O_3$ in the troposphere

Anyway, it seems that $O_3$ partial columns used here are heavily influenced, if not dominated by free tropospheric (FT) ozone located between, say, $2 - 8$ km height. Note that it would be useful to dispose of boundary layer height values allowing to state where the FT begins. This important, if not dominant, FT part for ozone needs to be clearly stated in the paper, while in the current version, it is suggested that the analysed product is representative for PBL $O_3$. If authors think that their product has not only some fractional, but dominant information from the PBL, then please explain and prove it much better. Note that this applies for ozone, for $NO_2$ and HCHO, the PBL sensitivity is stronger because both compounds are concentrated there.

If this suspected FT sensitivity is real, then the analysis needs to be taken into account in the following analysis. While PBL ozone can be strongly regionally controlled, FT ozone is prone to intercontinental or even hemispheric transport, including stratosphere − troposphere exchange. This needs to be addressed, and may be, the analysis completed or redirected. This point is really important to be addressed before

publication.

**Response:** Thanks very much for your useful and interesting comments. We hope the following statements can answer your questions.

1) Many scientists have proved that column technique (OMI, GOME, or airborne results) can be used to investigate $PO_3$ sensitivity, and the risky of the column technique was discussed in detail therein (Martin et al. 2004a; Duncan et al. 2010; Choi et al., 2012; Witte et al., 2011; Jin and Holloway, 2015; Mahajan et al., 2015; Schroeder et al., 2017; Jin et al., 2017). The $NO_2$ used in this study is the same as most previous studies, the sensitivity/resolution of FTS $O_3$ is close to that of OMI (Liu et al., 2010), the FTS HCHO is verified to be robust in troposphere in view of future satellite validation (Vigouroux et al., 2018). Thus, column technique used in this study is reasonable. We do acknowledge the paper by Schroeder et.al. (2017) which was published during the preparation of the manuscript. Schroeder et.al. (2017) question the usability of the column technique to infer PO3 sensitivity. However, this manuscript does take into account much of the criticism mentioned by Schroeder et.el (2017): we calculated the transition thresholds with the measurements in Hefei rather than straightly applied the thresholds estimated by either previous studies. The FTIR measurements have a much smaller footprint than the satellite measurements. Also we concentrate on measurements recorded during midday, when the mixing layer has largely been dissolved. And furthermore, the measurements are more sensitive to the lower parts of the troposphere, which can be inferred from the normalized AVK's. This reason is simply, that the AVK's show the sensitivity to the column, but the column per altitude decreases with altitude.

2) Column measurements sample a larger portion of the atmosphere, and thus their spatial coverage are larger than in situ measurements. So the photochemical scene disclosed by column measurement is larger than the in-situ measurement. The retrieval in section 3 shows that the tropospheric DOFS of $O_3$ is only slightly larger than 1.0 (similar to previous studies which use tropospheric OMI product). $O_3$ retrieval does show large sensitivity in the troposphere, but is not sufficient to divided into PBL part and FT part. Generally, this study reflects the mean photochemical

condition of the troposphere.

3) In photochemical reaction, $O_3$ production is more sensitive to VOCs if it is VOCs-limited and is more sensitive to $NO_x$ if it is $NO_x$ limited, and it exists a transition point near the threshold (Martin et al., 2004). The transported ozone (e.g., STE process), if not produced by photochemical reaction, may alter tropospheric ozone amount but would not sensitive to either VOCs or $NO_x$. Thus, it should not alter the photochemical regime estimated in this study. On the other hand, the selection of tropospheric limits 3 km below the tropopause minimized the influence of transport from stratosphere, the STE process.

**Related change:** We already included most of above explanation in the revised version.

Minor remarks:

Abstract :

Introduction:

"Briefly, VOCs first react with the hydroxyl radical (OH) to form a peroxy radical (HO2+ RO2) which increases the rate of catalytic cycling of NO to NO2. O3 is then produced by subsequent reactions between HO2or RO2 and NO that lead to radical propagation (via subsequent reformation of OH).

Please reformulate :

"Briefly, VOCs first react with the hydroxyl radical (OH) to form a peroxy radical (HO2+ RO2) which increases the rate of catalytic cycling of NO to NO2. O3 is then produced by photolysis of NO2. Subsequent reactions between HO2or RO2 and NO lead to radical propagation (via subsequent reformation of OH).

**Response:** This has been done.

Section 4.1:

"While it failed to determine the secular trend of tropospheric O3 column probably because the time series is much shorter than those in Gardiner et al. (2008),"

This sentence suggests that the $O_3$ trend needs to be positive, but for later years, this is not necessarily the case, especially in the free troposphere.

**Response:** As suggested by reviewer#2, this sentence has been revised to "The

analysis did not indicate a significant secular trend of tropospheric $O_3$ column probably because the time series is much shorter than those in Gardiner et al. (2008)……"

Section 4.2

"The direction of east origin air masses shifts from the southeast to northeast of Jiangsu Province, and that of local origin air masses shifts from the south to the northwest of Anhui province."

Please reformulate:

"The direction of air masses originating in the eastern sector shifts from the southeast to northeast of Jiangsu Province, and that of local air masses shifts from the south to the northwest of Anhui province."

**Response:** This has been done.

"In contrast, trajectories of local origin air masses in SON/DJF are 20.2% larger than the MAM/JJA ones, indicating a more significant contribution of the air pollution inside Anhui province in SON/DJF."

You mean, they are more frequent (instead of larger)?

**Response:** Yes, it is more frequent. In the revised version, the "larger" has been replaced by "more frequent".

5.1 Meteorological dependency

"The city downtown locates in eastern of the observation site and the majority of the Chinese population lives in the eastern part of China, easterly winds (direction less than 180˚) could generally transport more pollutants to the observe area than westerly winds (direction larger than 180˚), resulting in a higher $O_3$ level."

This again supposes, that the $O_3$ product is mostly sensitive top PBL $O_3$, which is clearly not proven. This also long range transport, not necessarily only from easterly regions, because trajectories can change directions, should contribute to enhanced $O_3$ columns.

**Response:** Since two referees worried/ puzzled about the wind data. We removed all wind data related panels and discussion. This sentence has been removed.

Section 5.2 :

"Pronounced tropospheric CO and NO$_2$ variations were observed but the seasonal cycles are not evident probably because of air pollution which is not constant over season or season dependent."

This is not clear. For NO2 a winter maximum is found. One reason is that time series are not complete enough especially in the later years.

**Response:** It has been changed to "Generally, tropospheric HCHO are higher and tropospheric NO$_2$ are lower in MAM/JJA than those in SON/DJF. Pronounced tropospheric CO was observed but the seasonal cycle is not evident probably because CO emission is not constant over season or season dependent."

"Since the sensitivity of PO3 to VOCs and NOx is different under different limitation regimes, the relative weaker overall correlations to HCHO (Figure 6 (b)) and NO2 (Figure 6 (c)) indicates that the O3 pollution in Hefei can neither be fully attributed to NOx pollution nor VOCs pollution."

If O$_3$ columns are representative for PBL, then weaker correlation of O$_3$ with NO$_2$ and HCHO are also explained by different lifetimes, hours to 1 day in summer for NO$_2$ and HCHO, several days to weeks for O$_3$. So older O$_3$ enhanced air masses easily loose trace of NO$_2$ or HCHO. Please add this explanation. If they are dominated by FT O$_3$, then a correlation can not be expected anyway. This seems to be the case.

**Response:** We have included this explanation in the revised version.

Figure 1:

In figure 1, how is the trend calculated? how are the points named 'resampled bootstrap" obtained. Please explain in the text.

**Response:** Since the method is already described in detail in Gardiner et al., 2008. We included this reference in the text to avoid repetition.

Figure 3 :

Monthly average wind speed of 0 does not make sense, as already stated by another referee. It is understood that the average of absolute wind speed is meant. If you cannot calculate it, please withdraw this figure. Also the figure on wind direction on wind direction is not easy to interprete. May be a solution is to calculate frequency distributions.

**Response:** Since two referees worried/ puzzled about the wind data. We removed all wind data related panels and discussion.

Figure 4:

If minute average FTS measurements are used, then there should be much more points? Or do you use time averages?

**Response:** The FTS retrievals within ± 30 min of OMI overpass time (13:30 local time (LT)) were averaged and used in this study.

**Related change:** We included this statement in the introduction (last paragraph).

Figure 5 and Figure 6: again, what is the measurement period one point corresponds to ?

**Response:** The FTS retrievals within ± 30 min of OMI overpass time (13:30 local time (LT)) were averaged and used in this study.

**Related change:** We included this statement in the introduction (last paragraph).

Supplement:

 "FigureS2. Averaging kernels(ppmv/ppmv)of O3, CO, and HCHO (color fine lines), and their area scaled by a factor of 0.2 (black bold line).They are deduced from the spectra recorded in Hefei on March 15, 2016 with a measured ILS."

Please clarify several points in the figures legend. Each curve is representative for 1 km ? What does the area curve exactly indicate? what does "ILS" stand for ?

**Response:** ILS stands for instrumental line shape. The atmosphere (0 − 120 km) is unevenly divided into 48 layers, and each curve corresponds to one layer which is not exactly for 1 km.

**Related change:** We have clarified several points in the figures legend.

 "FigureS3. Partial column averaging kernels(PAVK)(ppmv/ppmv) for O$_3$, CO, and HCHO retrievals. For all gases, large PAVKs in certain altitude range"

Are the PAVK's obtained by summing up the km wise AVK's ?

**Response:** Yes, PAVK's are obtained by summing up the concerned km AVK's.

 **(3) Detailed response to comments from referee #2:**

The authors have revised the manuscript quite extensively and have addressed many of the concerns raised by the reviewers. The manuscript now has a much clearer

message and storyline. The questionable comparisons to other observations and model simulations have been removed from the previous version. I think that this is now publishable in the ACP special edition, after a few mostly minor revisions.

**Response:** Thanks very much for your recommendation with respect to publish this paper in ACP. All glitches listed by you have been addressed. Please check details as below.

Generally, the English should be improved throughout the manuscript, especially in the later sections. I assume that much of this will be done in the copy-editing phase.

**Response:** We have tried our best to improve the language problem. I hope the ACP's copy and language editing service can fix the rest if we did not find out all of them.

line 22: I would delete "different". I think "and modelling" also needs to be deleted, because modelling is not really used anymore in the revised version.

**Response:** "different" and "and modelling" have been removed.

lines 30,31 (and 232,233; 452,453): I find this awkward. Why not state the June value 1.5e18? Given the variations and uncertainties 47.6% should probably be 50%. It would also be useful to give these column densities in Dobson Units (DU): 1.5e18 1/cm2 = 56 DU, 1.05e18/cm2 = 39 DU

**Response:** As your suggestion, these sentence have been changed to " Tropospheric $O_3$ columns in June are $1.55 \times 10^{18}$ molecules*cm$^{-2}$ (56 DU (Dobson Units)) and in December are $1.05 \times 10^{18}$ molecules*cm$^{-2}$ (39 DU). Tropospheric $O_3$ columns in June were ~ 50% higher than those in December."

lines 32 to 35: This sentence does not make sense to me. MAM/JJA is mainly influence by transport whereas MAM/JJA is mostly determined by photochemical production? Please rethink and reword.

**Response:** Has been changed to "Compared with SON/DJF season, the observed tropospheric $O_3$ levels in MAM/JJA are more influenced by transport of air masses from densely populated and industrialized areas, and the high $O_3$ level and variability in MAM/JJA is determined by the photochemical $O_3$ production."

line 40: % of what? Probably "of days" or "of cases". Change as appropriate.

**Response:** has been revised to "% of days"

line 42: delete "most of the". This has not been shown.

**Response:** "most of the" has been removed.

line 222: I would replace "failed to determine the" by something like "the analysis did not indicate a significant"

**Response:** has been replaced by " the analysis did not indicate a significant….. " .

line 229: delete "it shows that"

**Response:** has been done.

lines 237 to 240: Not really true. Izana also shows an early summer maximum in May, at a latitude not too different from Hefei. Somewhere, you should also state that the tropospheric columns at these stations are of the order of 0.7e18/cm2 to 1.1e18/cm2, around 30% lower than at Hefei.

**Response:** "The tropospheric columns at these stations are of the order of $0.7 \times 10^{18}$ molecules*$cm^{-2}$ to $1.1 \times 10^{18}$ molecules*$cm^{-2}$, around 30% lower than at Hefei." and "The results showed a maximum tropospheric $O_3$ column in spring at all these stations except at the high altitude stations Jungfraujoch and Izaña where it extended into early summer." have been included in the revised version.

lines 249 to 251: I find this sentence confusing and I suggest to omit it.

**Response:** This sentence has been removed.

line 262, and throughout section 4.2.: In most cases "air pollution" should be replaced by "air masses". The trajectory does not tell you anything about air pollution and its photo-chemical effects. It just tells you where the air masses (might) come from. Everything else is assumptions and plausibility.

**Response:** This has been done in the revised version.

line 276: Replace "dominate the contribution" by "contributes". Without a quantitative analysis, you do not know what dominates.

**Response:** This has been done in the revised version.

line 277: Replace "masses" by "trajectories".

**Response:** This has been done in the revised version.

line 282: replace "smaller" by "less frequent"

**Response:** This has been done in the revised version.

line 300: delete "which can be"

**Response:** This has been done in the revised version.

lines 308 to 318, Figs. 3 and 4: I am a bit worried/ puzzled about the wind data. Wind direction seems to be all over the place, and wind speeds seems to vary greatly between minutes and hours. Where do these short wind gusts come from? How representative is the local scale weather-station wind for the tropospheric FTIR columns? It might be much better to use larger scale winds, e.g. from the GDAS data? I think the wind data in Figs. 3 and 4 need careful checking, and it may be necessary to change or drop these panels and their discussion.

**Response:** Since two referees worried/ puzzled about the wind data. We removed all wind data related panels and discussion.

lines 339, 340: Something wrong with that sentence. Please fix.

**Response:** Has been revised to "Figure 5 shows time series of tropospheric CO, HCHO, and $NO_2$ columns that are coincident with $O_3$ counterparts……"

lines 368 to 385: I don't think any of this information is needed here, and I find it more confusing than helpful. I suggest to drop this text.

**Response:** All these information have been removed.

lines 414 to 423: So this study finds a VOC - NOx transition regime at HCHO/NOx ratios between 1.3 and 2.8, whereas previous (in situ based) studies found it at HCHO/NOx ratios between 1 and 2. I think this is pretty good agreement and could be emphasized. You point out that your study is not in-situ, and this could explain some differences. Could you please also add a statement about possible chemical differences?

**Response:** Duncan et al. (2010) concluded that $O_3$ production decreases with reductions in VOCs at column HCHO/$NO_2$ ratio < 1.0 and $NO_x$ at column HCHO/$NO_2$ ratio > 2.0; both $NO_x$ and VOCs reductions decrease $O_3$ production when column HCHO/$NO_2$ ratio lies in between 1.0 and 2.0. This means, HCHO/NOx ratios between 1 and 2 is also based on tropospheric column technique and not from in-situ. Regarding the possible chemical difference between in situ and column measurement can be found in previous studies like Choi et al., 2012; Witte et al., 2011; Jin and

Holloway, 2015; Mahajan et al., 2015; Jin et al., 2017 or Schroeder et al. 2017. Briefly, column measurements sample a larger portion of the atmosphere, and thus their spatial coverage are larger than in situ measurements. So the photochemical scene disclosed by column measurement is larger than the in-situ measurement.

**Response:** We have include this statement about possible chemical differences in section 5.3.2.

**(4) Detailed response to comments from referee #3:**

The paper documents retrievals of ozone, formaldehyde, and carbon monoxide using FTS observations in the central portion of eastern China over the 2014-2017 period. The paper also provides some reasonable analysis of the sources of the ozone values at this station. I read the revised version, and went back and looked over the original version. First, the authors have put in a nice effort into revising their manuscript and mainly responding to the critiques of the original manuscript. Second, the discussion on the retrievals, kernels, uncertainties is well done. I would rate my confidence in the data and retrievals to be quite high. Third, tropospheric chemistry in not my forte, but I'm skeptical of inferring information from column observations. Since ozone production is a non-linear function of VOCs and NOx, and since much of this is found in the boundary layer or strong plumes in the free troposphere, it is difficult to inter much information from column observations. Hence, I find some of the correlations shown in the manuscript to be not well founded or poor. Hence, the overall analysis is marginal in my "dynamicist" view. Nevertheless, this is interesting data from a highly polluted region that should be in the literature. Hence, I feel this should be published.

**Response:** Thanks very much for your recommendation with respect to publish this paper in ACP. Regarding your skeptical of inferring information from column observations, here are our explanations:

1) Column measurements sample a larger portion of the atmosphere, and thus their spatial coverage are larger than in situ measurements. So the photochemical scene disclosed by column measurement is larger than the in-situ measurement. The retrieval in section 3 shows that the tropospheric DOFS of $O_3$ is only slightly larger than 1.0 (similar to previous studies which use tropospheric OMI product). $O_3$

retrieval does show large sensitivity in the troposphere, but is not sufficient to divided into PBL part and FT part. Generally, this study reflects the mean photochemical condition of the troposphere.

2) Many scientists have proved that column technique (OMI, GOME, or airborne results) can be used to investigate $PO_3$ sensitivity, and the risky of the column technique was discussed in detail therein (Martin et al. 2004a; Duncan et al. 2010; Choi et al., 2012; Witte et al., 2011; Jin and Holloway, 2015; Mahajan et al., 2015; Schroeder et al., 2017; Jin et al., 2017). The $NO_2$ used in this study is the same as most previous studies, the sensitivity/resolution of FTS $O_3$ is close to that of OMI (Liu et al., 2010), the FTS HCHO is verified to be robust in troposphere in view of future satellite validation (Vigouroux et al., 2018). Thus, column technique used in this study is reasonable. We do acknowledge the paper by Schroeder et.al. (2017) which was published during the preparation of the manuscript. Schroeder et.al. (2017) question the usability of the column technique to infer PO3 sensitivity. However, this manuscript does take into account much of the criticism mentioned by Schroeder et.el (2017): we calculated the transition thresholds with the measurements in Hefei rather than straightly applied the thresholds estimated by either previous studies. The FTIR measurements have a much smaller footprint than the satellite measurements. Also we concentrate on measurements recorded during midday, when the mixing layer has largely been dissolved. And furthermore, the measurements are more sensitive to the lower parts of the troposphere, which can be inferred from the normalized AVK's. This reason is simply, that the AVK's show the sensitivity to the column, but the column per altitude decreases with altitude.

**In detail:** Over polluted areas, both HCHO and tropospheric $NO_2$ have vertical distributions that are heavily weighted toward the lower troposphere, indicating that tropospheric column measurements of these gases are fairly representative of near surface conditions. Many studies have taken advantage of these favorable vertical distributions to investigate surface emissions of NOx and VOCs from space (Boersma et al., 2009; Martin et al., 2004a; Millet et al., 2008; Streets et al., 2013). Martin et al. (2004a) and Duncan et al. (2010) used satellite measurements of column HCHO/$NO_2$

ratio to explore tropospheric $O_3$ sensitivities from space and disclosed that this diagnosis of $O_3$ production rate ($PO_3$) is consistent with previous finding of surface photochemistry. Witte et al. (2011) used a similar technique to estimate changes in $PO_3$ to the strict emission control measures (ECMs) during Beijing Summer Olympic Games period in 2008. Recent papers have applied the findings of Duncan et al. (2010) to observe $O_3$ sensitivity in other parts of the world (Choi et al., 2012; Witte et al., 2011; Jin and Holloway, 2015; Mahajan et al., 2015; Schroeder et al., 2017; Jin et al., 2017).

**Section 2: marked up file, as follows**

In briefly, we have moved all wind data related panels in Figs. 3 and 4, and the corresponding discussion in section 5.1. Other minor revisions responded to referees' comments also performed. We have tried our best to improve the language problem. The marked up file is as follow, please check the red underlined sentences for details:

[revised manuscript text omitted]